# Sulfide resorption during crustal ascent and degassing of oceanic plateau basalts

C.D.J. Reekie [1], F.E. Jenner[2], D.J. Smythe[3], E.H. Hauri[4], E.S. Bullock[5] & H.M. Williams[1]

Mantle plume-related magmas typically have higher chalcophile and siderophile element (CSE) contents than mid-ocean ridge basalts (MORB). These differences are often attributed to sulfide-under-saturation of plume-related melts. However, because of eruption-related degassing of sulfur (S) and the compositional, pressure, temperature and redox effects on S-solubility, understanding the magmatic behavior of S is challenging. Using CSE data for oceanic plateau basalts (OPB), which rarely degas S, we show that many OPB are sulfide-saturated. Differences in the timing of sulfide-saturation between individual OPB suites can be explained by pressure effects on sulfur solubility associated with ascent through over-thickened crust. Importantly, where S-degassing does occur, OPB have higher CSE contents than S-undegassed melts at similar stages of differentiation. This can be explained by resorption of earlier-formed sulfides, which might play an important role in enriching degassed melts in sulfide-compatible CSE and potentially contributes to anomalous enrichments of CSE in the crust.

[1] Department of Earth Sciences, University of Cambridge, Downing Street, Cambridge CB2 3EQ, UK. [2] School of Environment, Earth and Ecosystem Sciences, The Open University, Walton Hall, Milton Keynes MK7 6AA, UK. [3] Department of Earth Sciences, University of Oxford, South Parks Road, Oxford OX1 3AN, UK. [4] Department of Terrestrial Magnetism, Carnegie Institution of Washington, 5241 Broad Branch Road, Washington DC, 20015, USA. [5] Geophysical Laboratory, Carnegie Institution of Washington, 5251 Broad Branch Road, Washington DC, 20015, USA. Correspondence and requests for materials should be addressed to C.D.J.R. (email: cdjr2@cam.ac.uk)

Sulfur (S) plays an important role in geochemical cycling in magmatic systems. During mantle melting and crustal differentiation, sulfide phases control the distribution of S and sulfide-hosted chalcophile (sulfide-loving) and siderophile (iron-loving) elements (collectively termed CSE) between the mantle and various parts of the crust[1,2]. During eruption at low pressure (i.e., subaerial and shallow-marine), S commonly degasses, releasing $SO_2$ and $H_2S$ to the atmosphere, with potentially damaging environmental effects[3]. Well-constrained S-systematics are therefore important for assessing the environmental hazards associated with volcanic eruptions, understanding the formation of ore deposits and placing constraints on the chemical evolution of the mantle, crust and atmosphere[1–5].

When a melt becomes sulfide-saturated, sulfides form and sequester CSE from the melt[1,6]. Because S is often degassed, reconstructing the behavior of S prior to eruption typically relies on CSE systematics[7]. For example, Cu- and Ag-depletions during differentiation (Fig. 1a, b) have been used to demonstrate that MORB fractionate sulfides as they cool[1]. By contrast, increasing [Cu] and [Ag] (where square brackets denote concentration) during differentiation of MORB-type magmas erupted in proximity to known plume tracks (plume-influenced MORB; PI-MORB), may indicate that the melts are sulfide-under-saturated[8]. Sulfur behavior during magmatic processes can also be constrained by quantifying the maximum solubility of sulfur in silicate liquids (sulfur content, [S], at sulfide-saturation, $[S]_{SULF-SAT}$)[9–12]. $[S]_{SULF-SAT}$ increases with increasing temperature, [FeO] of the melt (over typical basaltic compositions) and $fO_2$, but decreases

with increasing pressure[9–12]. Sulfur solubility is therefore sensitive to processes such as magma cooling, fractional crystallization, magma ascent, magma chamber recharge and crustal contamination. Hence, complex processes, such as sulfide formation and later dissolution during ascent, might occur during crustal differentiation. However, isolating and assessing the effect of these variables is challenging. In addition, uncertainty regarding the [S] of the Earth's mantle, and hence the modal abundance of mantle sulfide, limits our understanding of how CSE are distributed between partial melts and their mantle residue during basalt petrogenesis[13–15].

Oceanic plateaus are igneous provinces comprised of large volumes (up to $5 \times 10^7$ km$^3$) of basaltic rock erupted over relatively short timescales[16,17]. They are considered to be derived from relatively short-lived mantle plumes[18,19]. Despite differences in their formation, the major and trace element systematics and $fO_2$ of oceanic plateau basalts (OPB) and MORB are similar[20]. Furthermore, unlike continental flood basalts, OPB are not contaminated by continental lithosphere. These aspects of OPB chemistry, coupled with clear differences in crustal thickness (up to 30 km; ref. [21]) relative to the oceanic crust (~6.5 km; ref. [22]) allow us to use OPB to isolate and assess the effects of degree of partial melting and pressure on $[S]_{SULF-SAT}$.

Here we present major, trace and volatile element data for quenched volcanic glasses from three Cretaceous oceanic plateaus (Ontong Java, Shatsky Rise, and Kerguelen), together with new CSE data (e.g., Se, Mo, Sn) for 439 MORB samples originally analyzed by Jenner and O'Neill[23] (see Methods). We show that

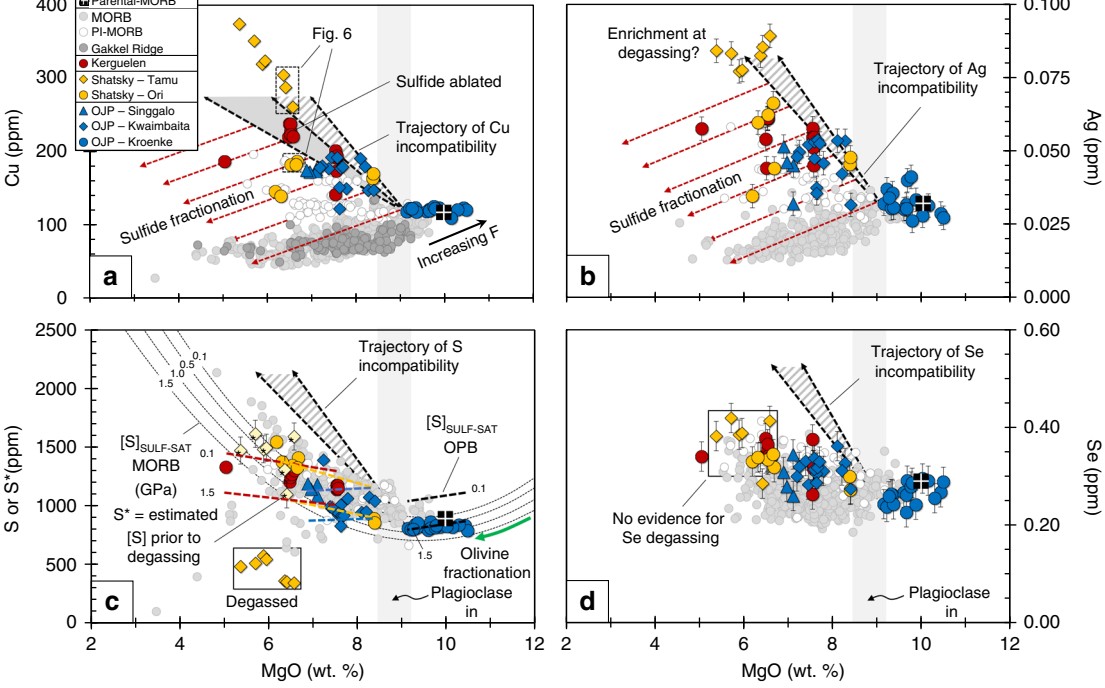

**Fig. 1** Chalcophile element variations during crustal differentiation of OPB and MORB. **a** [Cu], **b** [Ag], **c** [S] and **d** [Se] versus [MgO]. The primitive Kroenke magmas have compositions which are similar to parental-MORB and are offset to higher [Cu] at a given [MgO] than samples from the ultra-slow spreading Gakkel Ridge, which are an example of low-degree partial melts[50,51]. Trajectories of increasing [Cu], [Ag], [S] and [Se] in a sulfide-under-saturated melt (gray-dashed triangle) are modeled assuming that these elements are perfectly incompatible and thus have a bulk-D of ~0. An additional trajectory for Cu in a sulfide-under-saturated melt is modeled using available partition coefficients for Cu in silicate phases ($D_{Cu}$ values taken from ref. [39]), giving the gray field on **a**. Many of the OPB samples plot below the "incompatible (D = 0) trajectories", suggesting that the magmas are sulfide-saturated. Sulfide fractionation lines (red dashed lines, calculated using the slope for Cu and Ag in sulfide-saturated MORB magmas; ref. [1]) suggest that the proportion of sulfides fractionated from OPB and MORB are comparable once saturation is achieved. OPB magmas saturate in sulfide at lower [MgO] than MORB. Unlike S, Se is not degassed from the Tamu Massif samples. Except Se (see Methods), data for MORB, parental-MORB and the Gakkel Ridge are taken from Jenner and O'Neill[23], Jenner[1] and Gale et al.[50], respectively. Error bars for Se and Ag are given as the 1SD error on repeat analysis of the NWLSC Se and Ag standards (see Methods), with error bars on S* representing the propagated 1SD error of Se

most OPB saturate in sulfide during crustal differentiation. However, because OPB are generated by higher degrees of partial melting compared to MORB and consequently, have slightly higher [FeO], sulfide-saturation is achieved after more pronounced fractional crystallization compared to typical MORB. The effects of this slightly higher [FeO] compared to MORB are offset by pressure effects on $[S]_{SULF-SAT}$. Furthermore, differences in the timing of sulfide-saturation between different OPB suites can be attributed to differences in pressure of differentiation. Lastly, we demonstrate that most OPB do not degas S on eruption, which potentially limits their ability to cause environmental impacts such as those associated with eruption of continental flood basalts. However, in the rare cases where OPB do degas S, the drop in [S] of the melt appears to cause pre-existing sulfides to dissolve and results in the production of extremely metal enriched melts.

## Results

**Major and CSE systematics.** There is general consensus that OPB are generated by higher degrees of partial melting (~30%; ref. [24]) and fractionate at pressures equal to or greater than MORB[25–27] (see also Supplementary Information, SI). For example, primitive OPB are offset to slightly higher [$FeO_T$] at a given [MgO] compared to MORB, which is consistent with the expected partial melting trajectory[28]. Despite these differences, primitive (~10 wt. % MgO) OPB (i.e., Kroenke basalts, Ontong Java Plateau) have similar [Cu], [Ag], [S], and [Se] at a given [MgO] compared to average parental-MORB (Fig. 1). The inflection in [S] and major elements versus [MgO] trends (Fig. 1 and SI) suggests that OPB and MORB both saturate in plagioclase at ~9 wt.% MgO[19,27,29]. Although the OPB suites are not expected to be co-genetic, these systematics suggest that OPB magmas have similar liquid lines of descent to MORB (i.e., fractionated similar mineral assemblages and consequently, equilibrated at similar crustal pressures prior to eruption) between 9 and 6 wt.% MgO.

Together, the OPB suites show increasing [Cu] and [Ag] with decreasing [MgO] (Fig. 1) and have lower [S] at a given [$FeO_T$] compared to the MORB array (Fig. 2). However, most OPB have comparable Cu/Ag to global MORB (Fig. 3). The slightly lower [S] at a given [$FeO_T$] of ocean island basalts (e.g., Iceland) and OPB magmas (Ontong Java) relative to MORB are often attributed to these melts being sulfide-under-saturated and/or to S degassing[24,30–32]. However, sulfide minerals were routinely observed and ablated (see SI) during our analyses of OPB glasses (sample containing ablated sulfide shown in Fig. 1a) and have been documented in previous studies[33], demonstrating that OPB magmas reach sulfide-saturation. Furthermore, because Se is less prone to degassing than S during eruption[8,34], comparable [Se] at a given [MgO] between OPB and MORB, but the lower S/Se of the Tamu Massif samples compared to primitive mantle, indicate that only the Tamu Massif (Shatsky Rise) samples degassed S during eruption (Figs. 1 and 3). This finding indicates that, like MORB, OPB are typically erupted under a deep enough water column to prevent S degassing.

**H₂O and CO₂ systematics.** Like MORB[35,36], the OPB magmas display an increase in [$H_2O$] with progressive fractionation (Fig. 4a), indicating that the melts were $H_2O$-under-saturated during both crustal differentiation and eruption. Consequently, the range in [$H_2O$] versus [MgO] of the OPB suite follow the trend predicted for fractional crystallization (Fig. 4a), rather than showing evidence for $H_2O$ degassing. By contrast, with the exception of a few Kwaimbaita samples and the Kroenke basalts, the [$CO_2$] at a given [MgO] of the majority of OPB magmas plot below the predicted fractional crystallization trend (Fig. 4b). The

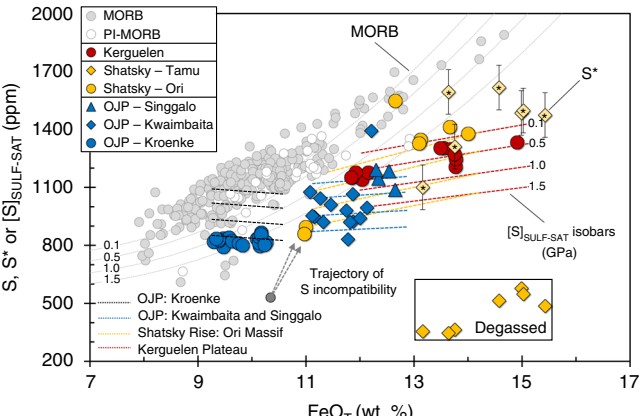

**Fig. 2** Pressure effects on sulfur solubility during crustal differentiation of OPB and MORB. [$FeO_T$] versus [S] for OPB samples and calculated [S] ([S*] = [Se]$_{Tamu}$ × S/Se$_{Ori}$) of Tamu Massif samples prior to degassing (yellow starred diamonds). [$FeO_T$] versus [S]$_{SULF-SAT}$ isobars (dashed thin lines) at four pressure intervals (0.1, 0.5, 1.0 and 1.5 GPa) are given for MORB and OPB suites and are calculated using the model of Smythe et al.[12]. MORB samples plot along the low pressure (~0.1 GPa) [S]$_{SULF-SAT}$ isobar because they are sulfide-saturated during low-pressure differentiation. OPB suites plot along modeled [S]$_{SULF-SAT}$ isobars at various pressures, suggesting that the different OPB suites may have stalled and differentiated at different pressures. Importantly, this also suggests that OPB magmas are sulfide-saturated, because they do not follow a steeper "D = 0" trajectory denoted by the gray-dashed arrow. Despite lying close to the 1.5 GPa isobar, the high [Pt] of the Kroenke samples (see SI) suggests that they were sulfide-under-saturated following ascent and consequently, eruption on the seafloor. The Tamu Massif samples are offset to low [S] at a given [$FeO_T$] compared to the Ori Massif samples, because they are degassed. Estimated [S] concentrations prior to degassing ([S*]) are scattered but indicate that the Tamu Massif may have been sulfide-saturated prior to degassing. Data for MORB are taken from Jenner and O'Neill[23]. Error bars on [S*] are given as the 1SD propagated error on repeat analysis of the NWLSC Se standards

OPB melts appear to reach $CO_2$ saturation between ~7.5 to 8.5 wt. % MgO and start degassing (Fig. 4b), which is consistent with the finding that $CO_2$ is more susceptible to degassing than $H_2O$ and S[37]. However, with the exception of the S-degassed Tamu Massif samples, partitioning of $H_2O$ and S into the exsolving $CO_2$-rich phase appears minimal.

## Discussion

To determine whether the CSE systematics of the OPB magmas preserve evidence for sulfide fractionation, we constrain the relative bulk-partitioning (bulk-D; calculated as the least-squares slope of log[M] versus [MgO], see Methods and SI) of all measured elements to provide a framework for understanding the behavior of the CSE. MORB slopes (i.e., the relative bulk-partitioning of trace elements in MORB; refs. [1,29]), plotted in order of increasing compatibility, are shown on Fig. 5. The comparable negative slopes of Rb, Cs, Th, Ba, Nb, Ta and W have been used to infer that these elements have a bulk-D of effectively 0 (incompatible) during differentiation[38]. Cu and Ag have near-identical positive slopes (i.e., are compatible), because both elements have similar sulfide-silicate melt partition coefficients ($D^{sulf/sil}$; refs. [1,6]).

Because most erupted MORB have fractionated plagioclase, which results in an increase in melt [$FeO_T$] and hence, [S]$_{SULF-SAT}$, OPB slopes are calculated using only the most-evolved

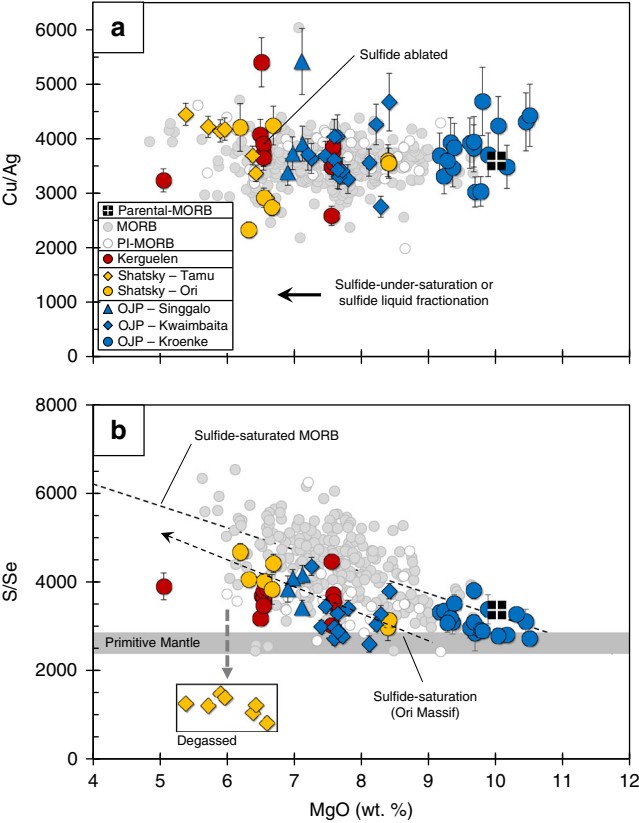

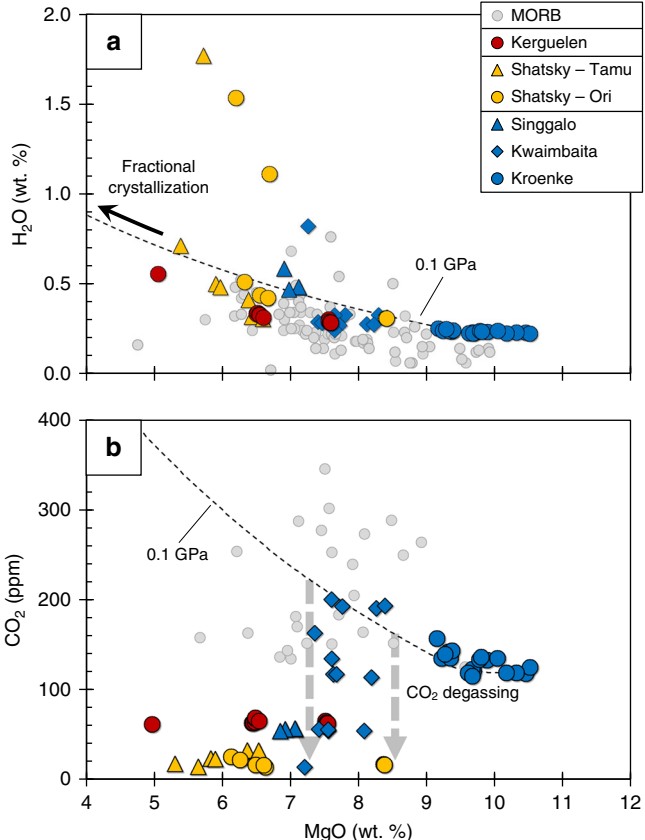

**Fig. 3** Cu/Ag and S/Se systematics during crustal differentiation of OPB and MORB. **a** Cu/Ag and **b** S/Se versus [MgO] of OPB and MORB. Because the $D^{sulf/sil}$ of Cu and Ag are similar (e.g., ref. [1]), both OPB and MORB have constant Cu/Ag regardless of whether the melts were sulfide-saturated or under-saturated. Sulfide-saturated MORB and the Ori Massif samples show an increase in S/Se with decreasing [MgO], consistent with sulfide fractionation. The Tamu Massif samples have considerably lower S/Se than other OPB samples and MORB, which is consistent with S-loss during degassing. Hence, the S/Se of the Ori Massif samples were used to estimate [S] of the Tamu Massif samples prior to degassing ([S*]). Some of the OPB are offset to lower S/Se than MORB and have comparable S/Se to primitive mantle estimates (ref. [13]), which might suggest that sulfide was exhausted from the OPB mantle source during melting. However, the Kroenke samples also have comparable S/Se to primitive MORB. Additionally, the lower S/Se are expected because the OPB magmas fractionated a smaller proportion of sulfide compared to MORB (i.e., MORB have lower [Cu] at a given [MgO] compared to OPB samples). Data for MORB are taken from Jenner and O'Neill[23], except Se data (see Methods). Error bars are given as the propagated 1SD Se error derived by repeat analysis of the NWLSC standards (see Methods)

**Fig. 4** H₂O and CO₂ systematics during crustal differentiation of OPB and MORB. **a** H₂O and **b** CO₂ versus [MgO] for OPB and MORB. Because S can strongly partition into a co-existing $H_2O-CO_2$ vapor phase (e.g. ref. [5]), it is important to constrain whether OPB were vapor-saturated or not during crustal differentiation. Like MORB, the H₂O contents of OPB increases with decreasing [MgO] and analyses closely overlap with a 0.1 GPa MELTS model (ref. [57]) of fractional crystallization of the most primitive (highest [MgO]) Kroenke sample (see accompanying supplementary note for further details regarding MELTS modeling). These systematics indicate that OPB were H₂O-under-saturated during differentiation. Silicate melts typically degas CO₂ before H₂O (e.g. ref. [37]) and this is consistent with the lower CO₂ of many of the OPB magmas compared to the trend predicted using MELTS modeling. Hence, with the exception of the S-degassed Tamu OPB, degassing of OPB was typically restricted to loss of CO₂. A few of the Kwaimbaita samples plot on the MELTS line, indicating that CO₂ saturation was reached at ~8.5−7.5 wt.% MgO and melts started to degas. MORB H₂O data are taken from Le Roux et al.[36] and Cottrell and Kelley[35]. MORB CO₂ data are taken from Le Roux et al.[36]. Assuming that the Kroenke samples are under-saturated and have not degassed H₂O or CO₂ and were generated by ~30% partial melting, we estimate that the OPB source mantle contained ≥700 ppm H₂O and ≥40 ppm CO₂

Kroenke samples (where plagioclase is inferred to join the liquidus, see Fig. 1 and SI) and the lower [MgO] OPB suites (except the S-degassed Tamu Massif) to allow comparison between MORB and OPB. A second slope (Fig. 5) is also calculated for each element following the above method, but excluding two samples that have extremely high incompatible trace element contents (e.g., Rb, Cs and Th) at a given [MgO] compared to the other OPB samples. The slopes calculated with and without the outliers (Fig. 5) are within error for most elements, but diverge slightly for some of the incompatible elements (e.g., Nb).

Most elements show comparable relative bulk-partitioning to MORB (Fig. 5), confirming that MORB and OPB fractionated similar mineral assemblages from 9 to 6 wt.% MgO. For example, Sr is more compatible (has a less negative slope) than Yb,

demonstrating that the OPB magmas, like MORB, fractionated plagioclase. Although Bi, Se, Cu, Ag, Re, Pt and Au have negative slopes during differentiation, the slopes of these highly chalcophile elements are significantly less negative (errors given on Fig. 5) than the "$D = 0$" elements (e.g., Rb) and are comparable to Sr (i.e., the CSE are being partitioned by a mineral during differentiation). Considering Se does not partition into silicate minerals, these CSE systematics are consistent with sulfide fractionation. Importantly, the slopes of Cu and Ag are statistically identical, implying that these elements have not been fractionated from each other during differentiation. Because Cu is more compatible in silicates, oxides and crystalline sulfides than Ag[1],

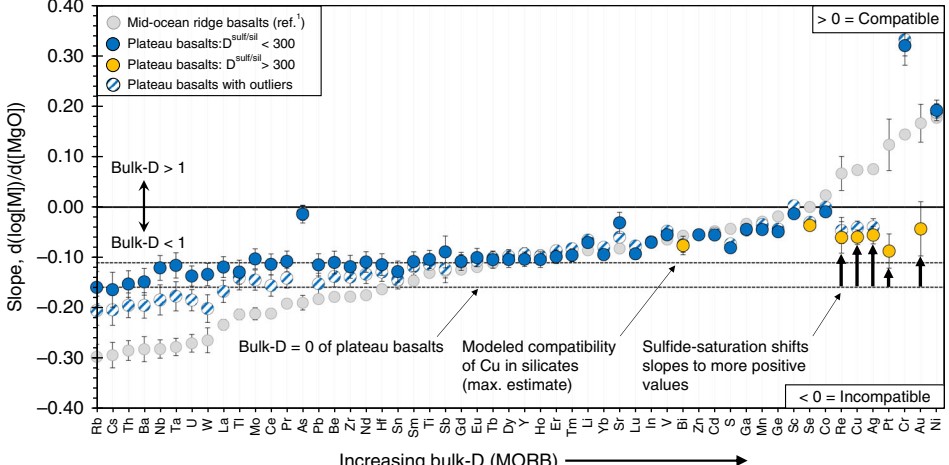

**Fig. 5** Relative bulk-partitioning of trace elements during crustal differentiation of OPB and MORB. Relative bulk-partitioning values (slopes; see Methods for details) for each element are calculated following the method of O'Neill and Jenner[29]. Two slopes are modeled for OPB (including and excluding two outliers which are enriched in highly incompatible elements (e.g., Rb, Cs and Th) at a given [MgO] compared to the other OPB samples) to provide a best-estimate of the range in compatibility of each element during crustal differentiation. Elements are ordered by increasing compatibility (i.e., more positive bulk-partitioning values) during MORB differentiation (MORB slopes taken from Jenner[1], except for Mo, Se, Sn, which were calculated using the new MORB data presented in Supplementary Table 4 and are listed in Supplementary Table 5; calculations performed without plume-influenced MORB). Elements with a slope < 0 are incompatible (i.e., preferentially remain in the melt phase during fractional crystallization; bulk-D < 1) whereas elements with a slope > 0 are compatible (i.e., preferentially incorporated into silicates, sulfide or oxides; bulk-D > 1). Together, CSE in OPB with a $D^{sulf/sil} > 300$ (yellow symbols) are bulk-incompatible (bulk-D < 1), but have more positive slopes than Rb, which has an estimated bulk-D of ~0 and is therefore taken as a proxy for the behavior of CSE in a sulfide-under-saturated melt. These offsets (black arrows) suggest that the melts reached sulfide-saturation during differentiation. Importantly, Cu is offset to a higher slope compared to a modeled bulk-D which assumes the behavior of Cu in the melt during differentiation is controlled solely by partitioning into silicate phases. Error bars are the standard error of the slope (see Methods)

fractionation of sulfide melt appears to be the cause of the comparable partitioning of Cu and Ag.

The slope of Rb, which has a bulk-D of ~0 during crustal differentiation, can be used to approximate the behavior of the CSE assuming that the melt was sulfide-under-saturated. Using both Rb slopes (i.e., slopes calculated with and without outliers) provides an estimate of the upper and lower limit of incompatible "D = 0" behavior of the CSE during differentiation of OPB magmas (Fig. 1). Because Cu can partition into silicate minerals, an additional slope is modeled for Cu (Figs. 1a and 5) in a sulfide-under-saturated melt using available partition coefficients[39]. Some of the high-[MgO] (>8 wt.%) samples from the Kwaimbaita and Ori Massif plot on or close to the "D = 0" trajectories (Fig. 1 and SI), implying that these samples were sulfide-under-saturated. Samples from the S-degassed Tamu Massif also plot on or close to the modeled sulfide-under-saturated trajectories for Cu, Ag, Pt and Au (Fig. 1 and SI).

Importantly, many of the OPB samples plot below the D = 0 trajectories, implying that these magmas were sulfide-saturated. The partitioning behavior of Cu and Ag after the melts reach sulfide-saturation can be approximated from the slopes of these elements in sulfide-saturated MORB (red dashed arrows on Fig. 1a, b), assuming for simplicity that there is no effect of temperature, pressure and melt composition on $D^{sulf/sil}$ (e.g. ref. [40]). Notably, some of the OPB and PI-MORB samples follow these sulfide fractionation trajectories. Hence, the Kwaimbaita and Ori Massif magmas appear to reach sulfide-saturation at ~7.5 wt.% MgO, which is supported by the trends to low [Pt] and [Au] and the increase in S/Se with further decreases in [MgO] (Fig. 3b and SI). Similarly, the high-MgO (~7.5 wt.% MgO) Kerguelen magmas show a range in [Cu], [Ag], [Se], S/Se and extremely low [Pt] and [Au] indicative of sulfide fractionation. However, the Cu systematics of lower [MgO] Kerguelen samples indicate sulfide-saturation at ≤6.5 wt.% MgO. Thus, OPB magmas

reach sulfide-saturation at a lower [MgO] than the ~9 wt.% MgO whereby most MORB achieve sulfide-saturation[1,34].

Given that the bulk-D of Cu increases significantly when a melt becomes sulfide-saturated during crustal differentiation, Rb/Cu and Cs/Cu ratios should be higher in sulfide-saturated melts compared to sulfide-under-saturated melts (see SI). However, Cu and Rb are fractionated from each other during mantle melting (e.g., in the presence of residual sulfide, Cu increases, whereas Rb decreases with increasing degrees of partial melting; refs. [41,42]). Hence, unlike "slopes" which are predominantly controlled by bulk-partitioning during crustal processes (e.g., fractional crystallization), chalcophile/lithophile element ratios are more sensitive to mantle processes. Instead, and as a further test of CSE systematics, we compare the compositions of the apparently sulfide-saturated (Ori Massif) and sulfide-under-saturated (Tamu Massif) samples with similar [MgO] from the Shatsky Rise. The differences in CSE between these two groups, expressed as [M]$_{Tamu}$/[M]$_{Ori}$ (groupings outlined on Fig. 1a) are plotted relative to $D^{sulf/sil}$ (Fig. 6). The non-CSE elements show a limited range in [M]$_{Tamu}$/[M]$_{Ori}$ of ~1, demonstrating the similarities in compositions between the two Massifs (Fig. 6). However, there is a strong correlation between [M]$_{Tamu}$/[M]$_{Ori}$ and $D^{sulf/sil}$ for elements with $D^{sulf/sil} > 100$, consistent with our interpretation that the Ori Massif samples are sulfide-saturated whereas the Tamu Massif samples were sulfide-under-saturated. Both Re and S are offset to low [M]$_{Tamu}$/[M]$_{Ori}$ compared to the overall trend, which is consistent with previous interpretations that Re, like S, is susceptible to degassing[43].

We have shown that the amount of crystallization required to reach sulfide-saturation differs between OPB and MORB. Given that the distribution of CSE between a mantle residue and partial melts is strongly controlled by mantle sulfide[41], it is important to constrain the potential for mantle processes to cause these differences. Modeling of the behavior of S during mantle processes is

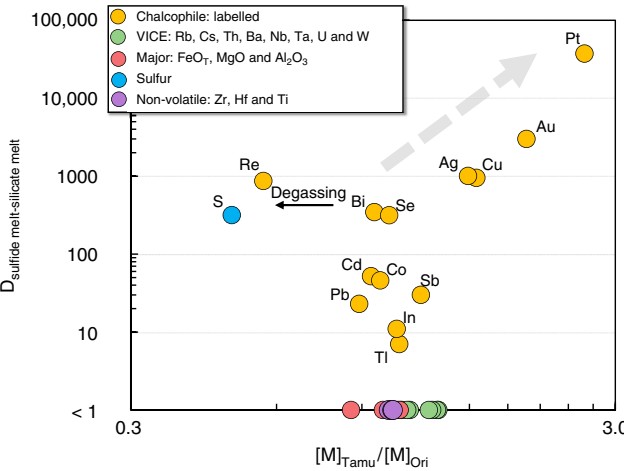

**Fig. 6** Ori-normalized CSE systematics in Shatsky Rise magmas as function of $D^{sulf/sil}$. Chalcophile element differences between two populations of Tamu Massif and Ori Massif samples ($[M]_{Tamu}/[M]_{Ori}$; see Fig. 1) are plotted with respect to their $D^{sulf/sil}$ (values taken from Jenner[1], and references therein). With the exception of Re, which shows evidence of degassing[43], highly chalcophile ($D^{sulf/sil} > 300$) elements show a strong correlation with $D^{sulf/sil}$, consistent with the interpretation that the Ori Massif samples are sulfide-saturated. Select incompatible, major and non-volatile elements are distributed around an $[M]_{Tamu}/[M]_{Ori}$ of 1, suggesting that the offset in chalcophile element contents between the Tamu and Ori Massifs at comparable [MgO] cannot be attributed to source heterogeneity and/or variations in the degree of partial melting. VICE = very incompatible elements (see ref. [38])

highly sensitive to parameters including melt composition, the [Cu] and [Ni] of mantle sulfide and the chosen $[S]_{SULF-SAT}$ model[41,42]. Together with the large range in estimates of the [S] of the primitive and depleted mantle[13–15] and therefore, the proportion of sulfide, considerable ambiguity remains regarding CSE systematics during mantle processes. For example, various arguments have been used to suggest that the MORB-source mantle contains residual sulfide and that clinopyroxene and sulfide are exhausted after the same degree of melting[1,11,44,45]. Alternatively, studies of platinum group element (PGE) systematics[46,47] and recent modeling using experimental constraints[41] have been used to argue that MORB are a mixture of low-degree sulfide-saturated (i.e., melting in the presence of residual sulfide) and high-degree sulfide-under-saturated partial melts (i.e., sulfide is exhausted during melting). However, most studies agree that the higher degrees of partial melting inferred for OPB melts and komatiites exhaust sulfide in the mantle source during melting[24,31,47].

Prior to sulfide exhaustion, the [S] of a partial melt is buffered at relatively constant [S] by $[S]_{SULF-SAT}$, the limit of which is dependent on the composition of the partial melt, pressure, temperature and $fO_2$. Following sulfide exhaustion, the [S] of the melt decreases significantly with increasing degrees of partial melting[41,42]. Hence, the delayed sulfide-saturation during differentiation might suggest that sulfide was exhausted from the OPB-source but not the MORB-source mantle during partial melting. Similarly, the [CaO] of partial melts increases with increasing degree of partial melting until the point of clinopyroxene exhaustion and then decreases with further melting[28]. In order to account for the differences in the degree of partial melting, but to achieve the similar to slightly higher [CaO] of OPB magmas compared to MORB (see SI), the OPB-source mantle needs to be more fertile than the depleted MORB-source mantle and

therefore, likely contained a higher proportion of both clinopyroxene and sulfide.

Sulfide-exhaustion is expected to offset OPB melts to lower Cu/Ag than MORB, because partition coefficients for Cu and Ag in silicate minerals are significantly different[1,48]. The comparable Cu/Ag of OPB and MORB (Fig. 3a) would therefore suggest that melting took place in the presence of residual sulfide melt in both settings. Alternatively, contrary to the modeling presented in Li and Audetat[48], these systematics may indicate that the differences in bulk-partitioning of Cu and Ag by silicates following sulfide exhaustion are not high enough to change the Cu/Ag of the melt with continued melting of the mantle. Because of difficulties in accurately measuring Ag in geological samples[49] and therefore, Ag partition coefficients of silicates, we instead rely on the use of well-constrained Cu systematics.

Prior to sulfide exhaustion, the [Cu] of partial melts increases with increasing degree of partial melting at low to moderate mantle potential temperatures (~1350−1450 °C), or shows a slight decrease at much higher mantle potential temperatures (~1650 °C)[41,42]. Following sulfide exhaustion, [Cu] of partial melts decreases rapidly with increased melting[41,42]. Samples from the slow-spreading Gakkel Ridge, taken as an example of low-degree partial melts[50,51], are offset to lower [Cu] at a given [MgO] compared to average global MORB and the Kroenke OPB (Fig. 1). Thus, like $FeO_T$, CaO and other major elements (see SI), the differences in [Cu] at a given [MgO] between Gakkel Ridge samples, global MORB and the Kroenke basalts appears to define a partial melting trajectory (see arrow on Fig. 1a). This trajectory is consistent with the systematics expected for increasing degrees of partial melting in the presence of residual sulfide.

Because OPB are generated by higher degrees of partial melting than MORB, if sulfide was residual in both mantle source regions, the Kroenke magmas should have higher [Cu] than parental-MORB. Alternatively, if sulfide was exhausted, the Kroenke magmas are expected to have lower [Cu] than parental-MORB, because the [CSE] of the melts would be diluted by continued melting. It is therefore surprising that the [Cu] of parental-MORB and the Kroenke magmas are indistinguishable. The similar [Cu] and comparable [CaO] between the Kroenke basalts and parental-MORB may indicate that melting of the "average" MORB-source and OPB-source mantle terminates at the point of clinopyroxene and sulfide exhaustion. Importantly, this suggests that OPB magmas are higher degree melts than MORB magmas simply because the mantle source was more fertile and contained more clinopyroxene and sulfide, compared to the depleted-MORB mantle. Additionally, we find that variations in the mantle source composition, fertility and/or the degree/depth of melting (e.g. variations in La/Sm and Gd/Yb) do not correlate with differences in the timing of sulfide-saturation between the various OPB suites and importantly, between OPB and MORB (see SI).

Sulfides are a major host of PGE in the mantle, which means the distribution of PGE between partial melts and the mantle residue is strongly linked to S-systematics[52]. Because of their extremely chalcophile nature ($D^{sulf-sil}$ from ~10,000 to >1 × 10⁶; ref. [53]), the low [PGE] in MORB are often used to infer that PGE were retained in residual sulfide during melting[46]. Conversely, the higher [PGE] in komatiites compared to MORB are used to infer that sulfide is exhausted from the mantle source during partial melting[54]. With the exception of Pt, Pd, Re and Au, primitive mantle-normalized[13] CSE abundances of the Kroenke basalts (Pt and Pd data from ref. [31]) are almost identical to parental-MORB (Fig. 7: note, because Ir, Ru and Rh are also compatible in spinel[55], these PGE are not discussed here). Notably, the Pt, Pd and Cu of komatiites are lower than Kroenke basalts (i.e., ~80 ppm Cu compared to ~120 ppm Cu, respectively), but the Pd/Cu and Pt/Pd are almost identical. Assuming sulfide was exhausted during

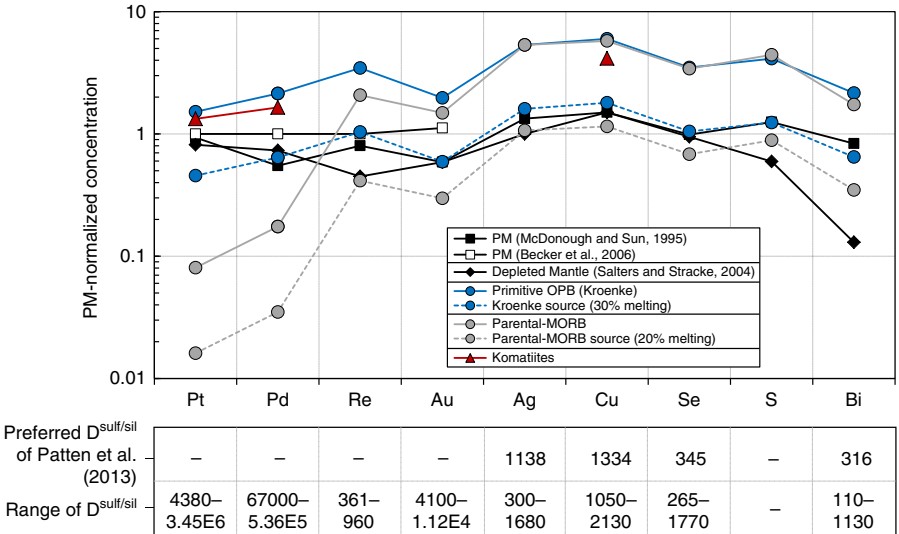

| | Pt | Pd | Re | Au | Ag | Cu | Se | S | Bi |
|---|---|---|---|---|---|---|---|---|---|
| Preferred $D^{sulf/sil}$ of Patten et al. (2013) | – | – | – | – | 1138 | 1334 | 345 | – | 316 |
| Range of $D^{sulf/sil}$ | 4380–3.45E6 | 67000–5.36E5 | 361–960 | 4100–1.12E4 | 300–1680 | 1050–2130 | 265–1770 | – | 110–1130 |

**Fig. 7** Primitive mantle-normalized PGE and CSE abundances of primitive OPB, MORB and komatiites. The CSE patterns of primitive OPB (Kroenke; blue line) and parental-MORB (orange line) are similar for Ag, Cu, Se, S and Bi. Together with their chondritic S/Se, this suggests that sulfide was exhausted from both the MORB-and OPB-source mantle and that melting terminated close to the point of sulfide-exhaustion. By contrast, komatiites are offset to lower primitive mantle-normalized Pt, Pd and Cu. Given that komatiites are generated by significantly high degrees of partial melting[54], these offsets suggest that the [CSE] of komatiites were diluted by continued melting in the absence of residual sulfide. Assuming that the CSE behaved perfectly incompatible after sulfide-exhaustion, estimates of the mantle source composition of OPB and MORB (dashed lines) are derived by assuming 30% and 20% degree of melting, respectively. The OPB mantle source estimate is well-matched to the primitive mantle estimates of McDonough and Sun[15], supporting our interpretation that sulfide was exhausted from the OPB mantle source. PGE data for OPB and MORB are taken from Chazey and Neal[31] and the compilation of Barnes et al.[46], respectively. Parental-MORB CSE estimates are taken from Jenner[1], except for Se, which is calculated from new MORB data presented in this study. PGE and Cu data for komatiites is taken from Puchtel and Humayan[54]. Sulfide-silicate partition coefficients are taken from Mungall and Brenan[53] (Pt, Pd and Au) and Patten et al.[6] (Re, Ag, Cu, Se and Bi). Data are normalized to the primitive mantle estimates of Palme and O'Neill[13]

melting and consequently, the CSE of OPB reflect their abundances in the mantle source region prior to 30% partial melting (i.e., [Cu] of average Kroenke multiplied by 0.3 gives the [Cu] of the mantle source region) gives a minimum estimate of the CSE composition of the Kroenke mantle (Fig. 7 and Supplementary Table 6). This composition is almost identical (except Pt) to the primitive mantle estimate of McDonough and Sun[15] and is comparable to the estimates of Palme and O'Neill[13]. Given the large differences in $D^{sulf/sil}$ between Pt, Pd, and Cu, these systematics indicate that (1) mantle melting to produce komatiites and the Kroenke basalts exhausted sulfide; (2) neither the komatiites nor the Kroenke basalts fractionated sulfide during differentiation and (3) the lower [Cu], [Pt], and [Pd] of komatiites compared to the Kroenke basalts are consistent with the expected dilution of [CSE] during mantle melting in the absence of sulfide (i.e., komatiites are higher degree partial melts than OPB).

The Ag-Cu-Se-S-Bi patterns of the Kroenke basalts show a close match to average parental-MORB. Additionally, the best-fit line through S/Se versus [MgO] of sulfide-saturated global MORB passes directly through the compositions of the Kroenke basalts, indicating that both OPB and parental-MORB have a S/Se that is chondritic (i.e., the same as the primitive mantle) prior to sulfide fractionation. These similarities indicate that neither the OPB nor the MORB-source mantle contain residual sulfide. Hence, the differences in Pt, Pd, Re and Au between MORB and the Kroenke basalts (Fig. 7 and SI) likely reflects partitioning during low-pressure differentiation, because unlike the Kroenke basalts, most erupted MORB are sulfide-saturated (see also SI). Assuming that sulfide was exhausted during melting and consequently, the CSE of MORB reflect their abundances in the source region prior to 20% partial melting (estimate from ref. [38]), we derive an estimate of the composition of the depleted mantle (Fig. 7 and

Supplementary Table 6). The Ag-Cu-Se-S-Bi patterns are parallel but offset to lower [CSE] compared to the more fertile OPB source mantle. Compared to previous estimates of the composition of the depleted mantle[14], our estimate is offset to higher [S] and [Bi] and lower [Au], [Cu] and [Se]. These differences can be attributed to the choices of ratios (S/Dy, Bi/Pb, Ir/Au, Cu/Sc, Se/V) used to derive the previous depleted mantle estimates, which are fractionated from each other during low-pressure differentiation of MORB[1].

Depending on the initial estimate of the [S] of the mantle, sulfide melt exhaustion is predicted to occur after ≤15% (ref. [48]). Considering the [CSE] of the melt will become diluted if melting continues past the point of sulfide exhaustion (i.e., dilution during an additional 15% partial melting to produce OPB parental magmas), our estimated [S] of the OPB-source mantle (247 ppm S) and the depleted MORB-source mantle (177 ppm) are strictly minimum estimates. However, considering our estimate of the [CSE] of the OPB mantle source is in good agreement with primitive mantle estimates (Fig. 7), we suggest that both the MORB and the OPB-source mantle only just exhausted sulfide during melting. Hence, previous modeling[41,42,48] might significantly underestimate the proportion of partial melting required to exhaust mantle sulfide.

Following our assessment of mantle processes and source fertility, the questions remain: why do OPB magmas typically reach sulfide-saturation after more pronounced crystallization than MORB, why does the crystallization interval differ between individual OPB suites and why do sulfide-saturated OPB have lower [S] at a given [FeO$_T$] compared to sulfide-saturated MORB? To address these questions, we use the Smythe et al.[12] model to calculate the [S]$_{SULF-SAT}$ of MORB at various pressure intervals across the measured range of major element concentrations, [Ni] and [Cu], assuming the composition of the sulfide in equilibrium

with the melt changes during fractionation. Melt temperatures were calculated from glass compositions using Eq. 14 from Putirka[56]. [S]$_{SULF-SAT}$ data points for MORB are fitted to a two-step polynomial to obtain [S]$_{SULF-SAT}$ versus [FeO$_T$] isobars (Fig. 2). Given difficulties in reproducing the MORB and OPB trends for some major elements using MELTS[57] and Petrolog[58] (e.g., CaO; see SI and ref. [59]), modeling results presented on Figs. 1 and 2 do not take into account the progressive delay in plagioclase-saturation (i.e., onset of FeO$_T$ enrichment) expected with an increasing pressure of differentiation. However, discussion and modeling of this effect is given in the SI for completeness.

MORB samples plot along a low-pressure (0.1 GPa) [S]$_{SULF-SAT}$ isobar (Fig. 2), consistent with the interpretation that MORB are sulfide-saturated at ~9 wt.% MgO during low-pressure differentiation[1,6]. Significant partial melting past sulfide exhaustion should dilute the [S] and [Cu] of the melt and consequently, increase the amount of crystallization required to achieve sulfide-saturation. Additionally, increasing [S]$_{SULF-SAT}$ with decreasing pressure during ascent from the mantle to the crust should push the melts further away from sulfide-saturation. Hence, the minimal crystallization required for MORB to achieve sulfide-saturation supports our conclusion that melting of the MORB- and OPB-source mantle terminates at approximately the same point as sulfide exhaustion.

Unlike MORB, where the potential range in differentiation pressures is small (i.e., the crust is thinner), plateau basalts could have fractionated at one or several pressures spanning from ~1 GPa (approximate depth to the base of the crust) to ~0.1 GPa prior to eruption[21,60]. Hence, [S]$_{SULF-SAT}$ isobars are modeled individually for each plateau (excluding the S-degassed Tamu Massif samples) using linear fits (Figs. 1 and 2). The modeled OPB isobars follow a trajectory that is sublinear to the MORB isobars and that diverges further with increasing [FeO$_T$] (Fig. 2). This can be attributed to compositional effects, because the [S]$_{SULF-SAT}$ of a silicate melt decreases with increasing [Cu][12,61] and the [Cu] of OPB at a given [MgO] and [FeO$_T$] are considerably higher than MORB, because of differences in the proportion of crystallization required to reach sulfide-saturation between OPB and MORB.

[S]$_{SULF-SAT}$ isobars modeled for the Kroenke basalts are higher at a given pressure compared to the MORB isobars (Fig. 1c). Given the similarities in Ag-Cu-Se-S-Bi and Ni between parental-MORB and the Kroenke OPB and the similar melt temperatures during crystallization, these differences can be attributed to the slightly higher [FeO$_T$] of the Kroenke magmas compared to parental-MORB (see SI). However, differences in [FeO$_T$] are offset by differences in crustal thickness between plateaus and the oceanic crust. For example, at mantle pressures (i.e., 1.5 GPa) and at the approximate base of the plateau (~1 GPa), the Kroenke magmas are close to sulfide-saturation (Figs. 1 and 2). However, if the Kroenke magmas ascended from the mantle to 0.1 GPa (i.e., the approximate pressure of crystallization of MORB), the Kroenke magmas would be ~270 ppm below their [S]$_{SULF-SAT}$. Hence, the high [MgO], [Pt], [Au] and [Re], together with the low [S] at a given [FeO$_T$] implies that the Kroenke magmas ascended from mantle depths to low-pressures and were subsequently erupted on the seafloor without sufficient crystallization to reach sulfide-saturation.

In contrast to the Kroenke samples, the [Cu] and [Ag] versus [MgO] systematics of the Kwaimbaita, Singgalo, Ori Massif and Kerguelen magmas provide evidence that the melts reached sulfide-saturation during differentiation (Fig. 1). Following plagioclase-saturation (i.e., samples with <9 wt.% MgO), if it is assumed that S had a bulk-D of ~0 prior to sulfide-saturation (i.e., increased at the same rate as incompatible elements, such as Rb),

the [S] of the melt will increase between ~350 and 490 ppm with every 1 wt.% decrease in [MgO] and the melt will quickly intersect the various [S]$_{SULF-SAT}$ isobars (Fig. 1). Therefore, assuming that the Kwaimbaita magmas evolved from a melt that was compositionally similar to the Kroenke magmas, if the Kwaimbaita magmas fractionated at 1 GPa the melts would have saturated at ~8.5 wt.% MgO and if the melts fractionated at 0.1 GPa the melts would have reached sulfide-saturation at ~8.2 wt.% MgO. These estimates are in close agreement with [Cu] versus [MgO] systematics which indicate that the Kwaimbaiata magmas became sulfide-saturated between ~8.4 and 8.1 wt.% MgO. Thus, the later saturation in sulfide of, for example, the low-[MgO] Kerguelen samples compared to the Kwaimbaita samples is consistent with the results of our modeling which indicates that the Kerguelen samples plot along the 0.5 GPa [S]$_{SULF-SAT}$ isobars whereas the Kwaimbaita samples plot predominantly below the 0.5 GPa isobars (Fig. 2). Because [S]$_{SULF-SAT}$ increases with decreasing pressure (Figs. 1 and 2), melts that reached sulfide-saturation at the base of a plateau will become sulfide-undersaturated during ascent and consequently, sulfides entrained in the ascending melt will dissolve. Staggered ascent (i.e., crystallization at more than one pressure) of the OPB magmas may therefore account for why magmas from a given suite (e.g., the two populations of Kwaimbaita and Kerguelen samples) appear to reach sulfide-saturation at different stages during their differentiation, and why the Kwaimbaita samples do not plot along a single [S]$_{SULF-SAT}$ isobar. Thus, the delayed onset of sulfide-saturation between OPB and MORB are a combination of the slightly higher initial [FeO$_T$] together with the likelihood that the melts may have repeatedly reached sulfide-saturation during differentiation but dissolved their sulfides during ascent.

Considering the relatively narrow crystallization interval required for OPB to reach sulfide-saturation, our [S]$_{SULF-SAT}$ modeling cannot explain why the Tamu Massif samples appear to be sulfide-under-saturated (e.g., Figs. 1 and 6). Using the [Se] of the Tamu Massif samples, and the average S/Se of the compositionally similar neighboring Ori Massif samples, gives estimated [S] of the Tamu Massif samples prior to eruption (S*; see ref. [34]). These values, while scattered, plot below the modeled "$D = 0$" trajectories (Fig. 1c) and either on or above the [S]$_{SULF-SAT}$ isobars (Figs. 1 and 2). If we assume that the Tamu Massif magmas differentiated without becoming sulfide-saturated, the [S] of the melt at 9 wt.% MgO would need to be extremely low (~400 ppm) to prevent sulfide-saturation during differentiation. This estimate is significantly lower than the measured [S] of the primitive Kroenke basalts. Additionally, the similarities in major and trace element compositions give no indication that the source mantle of the Tamu Massif samples was significantly depleted compared to the source of the other OPB. Alternatively, sulfide-saturation prior to degassing can potentially explain the high [CSE] of Tamu Massif samples relative to the Ori Massif at a given [MgO]. For example, degassing of S will cause melts to become sulfide-under-saturated, which is expected to result in destabilization and resorption of entrained sulfides and consequently, the release of sulfide-hosted CSE back into the melt. Therefore, sulfide resorption on ascent/and or during degassing (giving the illusion of "$D = 0$" behavior) could play an important role in enriching the melts in CSE beyond that expected of fractional crystallization.

We show that most OPB become sulfide-saturated during differentiation. The slightly higher [FeO$_T$] of OPB magmas relative to MORB contributes to OPB magmas reaching sulfide-saturation after more pronounced fractional crystallization compared to MORB. Additionally, we suggest that differences in the pressure of crystallization, rather than parental melt compositions, can account for different OPB suites reaching

sulfide-saturation at different stages of crystallization. Because the [CSE] of the S-degassed Tamu Massif samples extend beyond the crystallization range by which a melt is expected to reach sulfide-saturation, resorption of sulfides and release of the sequestered CSE back into the melt might have taken place. This could be an important and relatively unrecognized process that enriches degassed melts in CSE, with important implications for the distribution of CSE in plume-related magmas and potentially, the evolution of magmatic ore deposits.

## Methods

**Samples and analytical techniques**. Fresh volcanic glass samples analyzed in this study were recovered during Integrated Ocean Drilling Program expeditions 183 (Kerguelen; ref. [62]), 324 (Shatsky Rise; ref. [63]) and 192 (Ontong Java; ref. [64]), and Ocean Drilling Program expedition 130 (Ontong Java; ref. [65]). Volatile element analyses ($CO_2$, $H_2O$, S, F, P and Cl) were carried out on fresh volcanic glasses mounted in indium at the Department of Terrestrial Magnetism (DTM), Carnegie Institution of Washington, using a Cameca 6f ion microprobe and following the methods described in previous studies[66,67]. Volcanic glasses were also mounted in epoxy and major element analyses were undertaken using a JEOL JXA-8900 electron microprobe (EPMA) equipped with five wavelength dispersive spectrometers at the Smithsonian Institution, following techniques described in Jenner et al.[34]. Epoxy mounts were subsequently re-polished and analyzed for trace element contents by laser ablation inductively coupled mass spectrometry (LA-ICP-MS) using a Photon Machines G2 193 nm excimer laser system coupled to a Thermo iCap-Q ICP-MS at DTM. Analytical techniques for this method are described in detail in Jenner et al.[34]. In summary, NIST-SRM 612 was used as an external calibration with the exception of major elements whereby BCR-2G was used for external calibration. $^{29}Si$ was used for internal calibration of data. Four natural basaltic glasses recovered from the North-West Lau Spreading Centre (NWLSC), which have isotope dilution concentration measurements for Ag and Se[34,49], were used to make interference corrections during analysis of Ag and Se and as a check of data quality.

To ensure reproducibility and accuracy of the data (e.g., analyses of Sb and Bi were close to detection limits) and as a secondary check on the accuracy of interference corrections necessary for Se and Ag analysis (see ref. [34]), repeat analyses were undertaken at the School of Environment, Earth and Ecosystem Sciences at the Open University using a Photon Machines Analyte G2 193 nm excimer laser system coupled to an Agilent 8800 ICP-MS/MS. LA-ICP-MS operating conditions utilized a 10 Hz repetition rate; a beam diameter of 208 μm; ablation cell gas of $0.9 \, l \, min^{-1}$ He, with $5 \, ml \, min^{-1}$ $N_2$ added downstream of the sample cell to increase sensitivity. NIST-SRM 612 was used for external calibration and either $^{29}Si$ or $^{43}Ca$ was used for internal calibration of the data, depending on the analytical session. Reference material BCR-2G was analyzed repeatedly in each analytical session and across multiple analytical sessions at both DTM and the Open University. The relative standard deviation (precision) for average values is typically ≤5%. Average analyses of BCR-2G are typically within 5% of the preferred values given by Jenner and O'Neill[68] (Supplementary Table 1) with the exception of Ag. However, different chips of BCR-2G were found to have systematically different concentrations of chalcophile elements (e.g., Ag can vary from ~0.1 to 0.9 ppm), demonstrating that BCR-2G is not suitable for external calibration for many of the chalcophile and siderophile elements (see ref. [49] for further details).

To avoid argide, chloride, hydride and doubly charged REE interferences on $^{75}As$ and Se masses ($^{78}Se$ and $^{80}Se$), samples were analyzed using oxygen reaction ($0.1 \, ml \, min^{-1}$) mass-shift mode (i.e., $^{75}As$ was reacted onto mass 91; $^{78}Se$ was reacted onto mass 94; $^{80}Se$ was reacted onto mass 96; see ref. [69] for further details regarding the use of ICP-MS/MS analysis). Repeat analyses of the NWLSC reference materials were within 3% of isotope dilution analyses for Se. Additionally, mass-reacted data for Se in OPB shows a comparable range in values (Supplementary Table 2) to data analyzed on mass (i.e., using measured interference rates); demonstrating the accuracy of the interference corrections made on the DTM data (i.e., the DTM and Open University data show comparable slopes despite increasing Cl and REE with decreasing MgO, which would increase ClAr and $REE^{2+}$ interferences on $^{77}Se$ during analysis at DTM). However, the mass-reacted data are more scattered than the on-mass data, which is attributed to lower levels of sensitivity when using reaction cells[69]. Consequently, here we use the mass-reacted data simply as a demonstration of accuracy of the on-mass interference corrections. Compositional data for OPB glasses analyzed for this study are given in Supplementary Table 3.

Recent studies have noted an offset between the analyses of Mo/Ce and other CSE in the MORB array presented by Jenner and O'Neill[23] compared to subsequent datasets[70]. In addition, our recent work has demonstrated that the accuracy of in situ analyses of Se using LA-ICP-MS can be improved using the NWLSC glasses, which have high-precision isotope dilution, thiol cotton fiber separation and hydride generation ICP-MS data for Se, to measure and make interference corrections[34,49]. Hence, a subset (439) of the 616 MORB glass analyzed by Jenner and O'Neill[23] were re-analyzed using the new (more sensitive) ICP-MS at DTM. These analyses (Supplementary Table 4) showed improvements in the

accuracy of analyses of Mo, Se and Sn. For example, new data are in agreement with the data presented at high [MgO] (low Cl) by Jenner and O'Neill[23] but are progressively offset to lower [Se] at lower [MgO] (higher Cl). This divergence is consistent with a previous underestimation of the ClAr production rate (see Jenner et al.[34] for further details). Comparisons between the two datasets and BCR-2G analyses (Supplementary Table 4) show good agreement with analyses presented in Jenner and O'Neill[23] for Ag, Cd, In and Pb. Sb analyses of BCR-2G between the two datasets show good agreement between the two datasets, but MORB values diverge at low [Sb], reflecting analyses approaching the limit of detection of the technique, variability in [Sb] backgrounds and/or memory effects. BCR-2G and MORB data for Bi and Tl are consistently lower (by 11% for Bi and 17% for Tl in BCR-2G) in the recent data compared to the 2012 data, also indicative of analyses approaching the limit of detection, heterogeneity in BCR-2G for some elements and/or memory effects for "sticky" elements. Supplementary Table 4 provides a revised MORB dataset, highlighting the new, more accurate analyses for Se, Mo and Sn (referred to as "preferred") and new information values for the other elements (referred to as "replicate"). Updated values for the slopes and parental-MORB values for Se, Mo and Sn are given in Supplementary Table 5.

**Bulk-D estimation**. The relative bulk-distribution (bulk-D) of each trace element in the OPB "array" was calculated following the approach of O'Neill and Jenner[29] and Jenner[1], where the bulk-D (slope) is given by:

$$d(\log[M])/d[MgO],$$

where [M] refers to the concentration of element M in ppm and [MgO] is the concentration of MgO in wt.%. The standard error of the slope is given by:

$$s(\log[M])/\mathrm{sqrt}\left(\sum([MgO] - \overline{[MgO]})^2\right),$$

where $s(\log[M])$ is the standard error of regression and is calculated by:

$$\mathrm{sqrt}\left(\left(\sum\left(\log[M] - \log[MgO]_0 - b[MgO]\right)^2\right)\right),$$

where $\log[MgO]_0$ is the intercept at 0 wt.% MgO. A full description of this modeling is given by refs. [1,29], and an example slope calculation is given in the Supplementary Information. In the case of the highly chalcophile elements (i.e., high $D^{sulf/sil}$) Pt and Au, where sulfide-saturation leads to a considerable spread at any given [MgO], slopes are calculated using data for the most-evolved Kroenke and other suites with concentrations greater than 10 and 4 ppb, respectively (see SI). Consequently, the maximum possible slopes (including and excluding outlier samples, see main text) for each of these elements is given.

## Data availability

All compositional data obtained in this study, including analyses of standard reference materials, are included as supplementary data tables.

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

## Acknowledgements

We dedicate this research to the memory of our close friend and co-author Erik Hauri. We thank the International Ocean Discovery Program (IODP) for providing samples and the Australia−New Zealand IODP Consortium (ANZIC) for providing Legacy/Special Analytical Funding to F.E.J. and E.H.H. for this study. ANZIC is supported by the Australian Government through the Australian Research Council's LIEF funding scheme (LE0882854) and the Australian and New Zealand consortium of universities and government agencies. F.E.J. and H.M.W. acknowledge funding from the NERC "SoS Tellurium and Selenium Cycling and Supply (TeASe)" consortium grant (NE/M010848/1) and NERC "Deep Volatiles" consortium grant (NE/M000427/1), F.E.J. also acknowledges funding from the NERC "From Arc Magmas to Ore Systems" (FAMOS) consortium grant (NE/P017045/1) and H.M.W. from an ERC Starting Grant (306655 "Habitable Planet"). C.D.J.R. acknowledges a Ph.D. studentship from TeASe and support from the Cambridge Commonwealth and European Trust. We thank Hugh O'Neill and Bernie Wood for insightful discussions, Sam Hammond for maintenance of the Open University LA-ICP-MS facility and Jianhua Wang for expert assistance and care of the Carnegie SIMS lab.

## Author contributions

F.E.J. wrote the IODP grant and conceived the project. F.E.J., E.H.H., E.S.B. and C.D.J.R. analyzed the samples and processed the data and C.D.J.R., F.E.J., D.J.S. and H.M.W. modeled, interpreted and considered the implications of the data. C.D.J.R. and F.E.J. wrote the initial manuscript with contributions from H.M.W. All authors contributed to final editing of the manuscript.

## Additional information

**Competing interests:** The authors declare no competing interests.

