## [Peer Review File · Nature Communications]

Reviewer #1 (Remarks to the Author):

The paper reports a set of high quality analytical data gathered on several oceanic plateau basalts, focusing on the role of sulfur/sulfide on the behavior of so-called CSE. The paper explores in detail the contrasted patterns of CSE behavior between MORBs and plume related basalts. It concludes to an essential role of pressure in determining the timing of sulfide saturation, which differs between the two contexts because of differences in crustal thickness. This is a well and clearly written contribution.

The main weakness in the manuscript is the lack of consideration given to COHS gas saturation on S systematics. It is true that the OPB are for the most part very similar to MORBs. They however display a higher H₂O content. This point warrants further discussion. A higher H₂O content means that OPB will reach fluid saturation earlier than MORBs. A higher H₂O will also delay plagioclase saturation, hence affects Fe enrichment during fractionation, and in turn sulfide saturation (assuming that H₂O has no effect on sulfide saturation in silicate melts). It may be that the analysed H₂O contents, even if higher than those of MORBS, have no practical effect on the modelled parameters. I recommend however the authors to demonstrate this point by carrying out appropriate MELTs modelling with hydrous conditions. This would considerably strengthen the paper.

If the recovered glasses used in this study were all fluid unsaturated, this would be an important aspect that would merit to be highlighted as well (but I could not find the pressure of sample collection in the accompanying excel spreadsheet), since they would constrain the volatile content of the mantle source of OPB.

Apart from the above criticism, the paper applies standard geochemical reasoning and ends up with an interesting conclusion that deserves being published in a journal like Nature Communications. The geochemical data base provided will be helpful anyway to the geochemical community interested in constraining volatile behavior in magmas worldwide.

Bruno Scaillet

Reviewer #2 (Remarks to the Author):

The manuscript from Reekie et al presents new major, trace and volatile element data of volcanic glasses from three cretaceous oceanic plateaus. Based on the geochemistry systematics of S, chalcophile elements, siderophile elements and other trace elements versus major elements, the authors conclude that 1) given the higher chalcophile elements compared to MORBs, differentiated OPBs are sulfide saturated; 2) primitive melts parental to OPBs are sulfide saturated during partial melting; 3) chalcophile elements are more enriched in OPBs with significant sulfur degassing by sulfide resorption process. The topic is interesting to wide fields, and the method of considering partitioning of all the trace elements to arrive these conclusions is new. However, at current stage, I have some reservations regarding some of the assumptions, and modeling taken in the manuscript.

My biggest concern comes to the conclusion that melts parental to OPBs are sulfide saturated during partial melting. This conclusion seems to be simply based on the observation of the comparable Cu/Ag and chalcophile element abundances of OPBs and MORBs (Figure 3), assuming the primitive MORBs are sulfide saturated. First of all, in Figure 3b, some Cu/Ag ratios from Kroenke with high MgO (>10wt.%) are actually slightly higher than the values from MORBs at similar MgO concentration. Secondly, there are previous studies on S, PGEs and Au in MORBs through analytical geochemistry and modeling arguing that MORBs are mixture of high degree sulfide undersaturated melts and low degree sulfide saturated melts (Barnes et al., 2015; Ding and Dasgupta, 2017; Rehkämper et al., 1999) instead of pure sulfide saturated melts. If this is the case, the comparable Cu/Ag ratio of OPBs and MORBs cannot lead to the conclusion that OPBs are sulfide saturated during mantle melting. The last but not the least, if the primitive OPBs with ~900 ppm S derived by ~30% degree of melting is indeed in equilibrium with sulfide in its mantle source, as the authors suggest, it indicates > 270 ppm S present in its mantle source, which is much higher than most of the current estimate of S abundances in the upper mantle. This would be a very interesting conclusion showing huge sulfide heterogeneity in the Earth mantle. However, it seems not convincing enough to come up with this conclusion solely based on limited numbers of high MgO samples from a single sample place (Kroenke). Since the fate of sulfide during partial melting and the partitioning of chalcophile elements (Cu, Ag) are closely related to the pressure, temperature, degree of melting, and melt composition. I would recommend the authors to conduct some forward calculations along the melting path at the appropriate mantle potential temperature related to OPBs modelling the change of S, Cu and Ag in the partial melts. And if indeed the Cu/Ag ratio can be reproduced with an elevated sulfur concentration in the mantle, it would be great for the authors to also compile chalcophile element data from some other locations which are plume related and derived by high degree melting to check if such sulfur enrichment is just a local effect, or is a common phenomenon.

Another problem I have with the current manuscript is that even though all the new OPBs data cover a wide range of MgO from >10 wt.% to 5 wt.%, samples from individual OPB suite are either within small MgO range or with quite distinct MgO concentration. Since these different OPB suites are not co-genetic, more clarification and justification is needed to show that samples from each suite are indeed representative for the whole differentiation trend. For example, the conclusion that the OPBs are sulfide-saturated through differentiation mainly results from the slightly negative slopes of the highly chalcophile elements. However, it is not clear only through the method part that whether the calculated slopes in Figure 4 are representative of the whole differentiation range between 10-5 wt.% MgO for all the OPB suites, or are derived from a certain OPB suite at a certain MgO range. Since one of the key discussions of this manuscript is that why OPBs generally hit sulfide-saturation

later than MORBs, it is important to clarify for individual OPB suite at what MgO content the basalts become sulfide saturated during crystallization.

More details comments:

Line 69-71: This sentence seems to be misleading. At higher pressure, sulfide solubility would be lower, then one would expect that it takes less crystallization to reach sulfide saturation rather than more pronounced crystallization.

Line 83: what is calculated? How?

Line 91: If one looks at Cu and Ag systematics for individual suites in Figure 1, only samples from Tamu suite show increasing trend.

Line 92: In Figure 2, the sulfur concentration of all the OPBs data is about 100-200 ppm lower than S in the MORBs. Therefore, the word "significant" seems to be overused.

Line 174-175: How much S would be present if this is the case? How would the estimate compare to the S abundances of other mantle domains producing high-degree mantle melts, e?

Line 217-218: If the Kroenke primitive melts are also sulfide saturated as MORBs, why would the [Re], [Pt] and [Au] be the closest estimates of the unmodified mantle melts?

Reviewer #3 (Remarks to the Author):

The main point of the article is that for a given [MgO], OPB have lower CSE contents than would be expected if these elements were acting perfectly incompatibly (or close to it in the case of copper). This suggests that they must be sulfide saturated, although this happens later than in MORB, leading to and enrichment in CSE. The article suggests that the later saturation is a reflection of a larger pressure difference (and therefore sulfur solubility difference) between melting and subsequent differentiation of melts. Finally they suggest that degassed OPB samples have higher CSE related to resorption of sulfides.

My main criticisms are related to the lack of uncertainties. There don't appear to be error bars on any of the geochemical plots, and uncertainties on individual analyses would be useful in the supplementary table. This is particularly important for plots of the Ag and Se content, where uncertainties are likely high, and for the ratios in Figure 3 (particularly 3a, since Se and S data come from different sources with individual uncertainties).

To some extent this affects the conclusions. Firstly the authors suggest that Tamu samples must have been sulfide saturated because S^* values are high and that apparently $D=0$ behavior is caused by resorption of sulfides. However S^* values are calculated based on S/Se ratios and the Tamu S contents, both of which have uncertainties which must be propagated through and appropriate error bars for S^* need to be plotted. This conclusion is also based on the assumption that Ori S/Se ratio are a good estimate for Tamu which may be true, although there appears to be significant spread in the S/Se ratios at other OPB in this study. It would be interesting to see a range of S^* values based on the range of S/Se ratios.

Secondly the assertion that sulfide exhaustion can't explain lower S contents and explain later sulfide saturation is based on the fact that Cu/Ag ratios aren't significantly offset from MORB values. It would be really useful to see a back of envelope calculation here. Authors could calculate roughly what decrease in the sulfur contents of primary melts would be needed (maybe a few 100 ppm from the FeOt vs S plot) to explain their results. They could then calculate how much partial melting would have to occur after sulfide exhaustion to get these lower S contents, and what the expected difference in the Cu/Ag ratio would be, this could then be added to Figure 3b.

In figure 2 it wasn't clear why the modeled [S]sulf-sat isobars for plotted for OPB's and for MORB had such a different shape. The text suggests the OPB isobars are linear fits but it wasn't clear on what data they were linear fits to.

Finally I think it would be useful to have some of the plots used to calculate slopes for bulk-D values in the supplementary data. Looking at the spread in the MgO vs Ag data (and with associated uncertainties on the Ag values) it seems like it should be difficult to produce a fit which is statistically different from the trajectory of incompatibility.

I think that the conclusion that OPB are sulfide saturated, but that this occurs later than in MORB is likely valid, based on the data presented in Figure 1. The arguments made for the other claims in the paper make sense but are hard to asses without uncertainties on the data.

Reviewer Comments

Editor:

Your manuscript entitled "Pressure effects complicate our understanding of Earth's sulfur cycle" has now been seen by 3 referees, whose comments are appended below. You will see from their comments copied below that while they find your work of considerable potential interest, they have raised quite substantial concerns that must be addressed. In light of these comments, we cannot accept the manuscript for publication, but would be interested in considering a revised version that addresses these serious concerns.

The reviewers have made a number of useful suggestions on how to improve your manuscript and I would advise that you follow these.

We greatly appreciate the editor's and reviewer's comments and suggestions. We have revised our manuscript where suggested. Reviewer's comments and concerns are addressed in the responses below.

1) In particular, we expect as a matter of course that all uncertainty analysis is done and that all error bars are present on figures when required.

We have included error bars on figures, where requested, in addition to the uncertainty data initially given in Methods.

Reviewer #1: (Bruno Scaillet)

The paper reports a set of high quality analytical data gathered on several oceanic plateau basalts, focusing on the role of sulfur/sulfide on the behavior of so-called CSE. The paper explores in detail the contrasted patterns of CSE behavior between MORBs and plume related basalts. It concludes to an essential role of pressure in determining the timing of sulfide saturation, which differs between the two contexts because of differences in crustal thickness. This is a well and clearly written contribution.

Major comments

1) The main weakness in the manuscript is the lack of consideration given to COHS gas saturation on S systematics. It is true that the OPB are for the most part very similar to MORBs. They however display a higher H₂O content. This point warrants further discussion. A higher H₂O content means that OPB will reach fluid saturation earlier than MORBs. A higher H₂O will also delay plagioclase saturation, hence affects Fe enrichment during fractionation, and in turn sulfide saturation (assuming that H₂O has no effect on sulfide saturation in silicate melts). It may be that the analysed H₂O contents, even if higher than those of MORBS, have no practical effect on the modelled parameters. I recommend however the authors to demonstrate this point by carrying out appropriate MELTs modelling with hydrous conditions. This would considerably strengthen the paper.

All MELTS modelling presented in our initial submission was undertaken at hydrous conditions using the measured H₂O and CO₂ contents of the samples. We are in agreement that these volatiles can have important implications for S-systematics and following the reviewer's advice, we have revised our manuscript accordingly to now include a discussion of these points in the main text (lines 118-129, pages 5-6). We also now include a figure in the main text (Fig. 4) that shows H₂O and CO₂ systematics versus MgO of the OPB samples compared to the original MELTS modelling (i.e., MELTS modelling was previously only presented in the Supplementary Information with regards to the discussion about major element systematics).

In summary, oceanic plateau basalts (OPB) show an increase in [H₂O] with decreasing [MgO] and closely follow the predicted fractional crystallisation trend using MELTS (Fig. 4a), indicating that the melts were not saturated in H₂O. As pointed out by the reviewer, the higher [H₂O] of OPB relative to MORB might have the effect of suppressing plagioclase saturation to lower [MgO], with important implications for [S]_{SULF-SAT}. However, the inflection in the trends of [FeO_T] and [Al₂O₃] versus [MgO] (SI Fig. 1) suggest that OPB attain plagioclase saturation across a similar [MgO] range to MORB (~9 wt. %) (as discussed in lines 100-102, pages 4-5). Most OPB samples plot below the predicted MELTS fractional crystallisation trend for [CO₂] versus [MgO] and appear to reach CO₂-saturation between ~7.5 and ~8.5 wt. % MgO. Hence, with the exception of the Tamu Massif samples, partitioning of H₂O and S into the exsolving volatile (CO₂-dominated) phase appears minimal.

2) If the recovered glasses used in this study were all fluid unsaturated, this would be an important aspect that would merit to be highlighted as well (but I could not find the pressure of sample collection in the accompanying excel spreadsheet), since they would constrain the volatile content of the mantle source of OPB.

Reviewer 1 is correct to suggest that if the Kroenke were fluid-under-saturated, they would be important samples for constraining the volatile element concentrations of the OPB mantle source. We initially avoided discussion of this point, as we are aware that another group is doing work on the volatile contents of OPB magmas. However, to avoid this being viewed as a weakness to the manuscript we have now briefly discussed these points (Fig. 4 caption) by including estimates of the H₂O and CO₂ content of the OPB mantle source. These estimates are derived by assuming that these elements behave perfectly incompatible during melting and that the basalts were generated by a 30 % degree of partial melting.

Because of their size and age, oceanic plateaus have experienced varying amounts of post-emplacement subsidence (e.g., Roberge et al., 2005). Therefore, the depth of sample collection cannot be used to constrain the pressure (depth) at which the lavas were erupted on the sea floor, and consequently whether they were fluid-under-saturated at eruption. To avoid this difficulty, we chose to compare the samples to fluid-under-saturated trends predicted by MELTS modelling.

Apart from the above criticism, the paper applies standard geochemical reasoning and ends up with an interesting conclusion that deserves being published in a journal like Nature Communications. The geochemical data base provided will be helpful anyway to the geochemical community interested in constraining volatile behavior in magmas worldwide.

We thank this reviewer for their supportive and constructive comments and we are extremely pleased that they consider that our work 'deserves being published in a journal like Nature Communications'.

Reviewer #2: (Anonymous)

The manuscript from Reekie et al presents new major, trace and volatile element data of volcanic glasses from three cretaceous oceanic plateaus. Based on the geochemistry systematics of S, chalcophile elements, siderophile elements and other trace elements versus major elements, the authors conclude that 1) given the higher chalcophile elements compared to MORBs, differentiated OPBs are sulfide saturated; 2) primitive melts parental to OPBs are sulfide saturated during partial melting; 3) chalcophile elements are more enriched in OPBs with significant sulfur degassing by sulfide resorption process. The topic is interesting to wide fields, and the method of considering partitioning of all the trace elements to arrive these conclusions is new. However, at current stage, I have some reservations regarding some of the assumptions, and modeling taken in the manuscript.

Major comments

1) My biggest concern comes to the conclusion that melts parental to OPBs are sulfide saturated during partial melting. This conclusion seems to be simply based on the observation of the comparable Cu/Ag and chalcophile element abundances of OPBs and MORBs (Figure 3), assuming the primitive MORBs are sulfide saturated. First of all, in Figure 3b, some Cu/Ag ratios from Kroenke with high MgO (>10wt.%) are actually slightly higher than the values from MORBs at similar MgO concentration.

Except for two samples with high Cu/Ag (one Kerguelen and one Kwaimbaita sample), OPB show similar variability to MORB, within error, on a plot of Cu/Ag versus [MgO]. Because the modelling of Li and Audétat (2012) predicts that Cu/Ag will trend to ~ 0 once sulfide is exhausted from the mantle source, we previously argued that the similar Cu/Ag of OPB and MORB indicates that melting in both settings took place in the presence of residual sulfide. However, we now emphasise that our data could alternatively 'indicate that the differences in bulk partitioning of Cu and Ag by silicates following sulfide exhaustion are not high enough to change the Cu/Ag of the melt with continued melting of the mantle' (lines 237-239, page 10). Further discussion of whether the OPB melts were sulfide-saturated or not during partial melting is given below.

2) Secondly, there are previous studies on S, PGEs and Au in MORBs through analytical geochemistry and modeling arguing that MORBs are mixture of high degree sulfide undersaturated melts and low degree sulfide saturated melts (Barnes et al., 2015; Ding and Dasgupta, 2017; Rehkämper et al., 1999) instead of pure sulfide saturated melts. If this is the case, the comparable Cu/Ag ratio of OPBs and MORBs cannot lead to the conclusion that OPBs are sulfide saturated during mantle melting.

Following the reviewer's advice, we have extended our discussion (i.e., beyond the use of Cu/Ag systematics) of whether OPB were sulfide-saturated or not during mantle melting (lines

242-315, pages 10-13). Embracing the reviewer's criticism has strengthened our discussion considerably.

To avoid repetition of previous modelling (i.e., modelling suggested in Reviewer 2's comments, below), we have discussed the models of Ding and Dasgupta (2018, 2017), which are based on experimental and thermodynamic constraints, in our revised manuscript. In summary, the modelling of Ding and Dasgupta (2018; their Figure 5) demonstrates that, at mantle potential temperatures relevant to this work (1350-1450°C), the [Cu] of partial melts in equilibrium with mantle sulfide increases with an increasing degree of partial melting, whereas [S] remains relatively constant. By contrast, when sulfide is exhausted, the [Cu] and [S] of a partial melt is predicted to decrease significantly with continued melting. Therefore, the comparable [Cu] of parental-MORB and primitive OPB (Kroenke), which are higher than the [Cu] of the slow-spreading Gakkel Ridge (example of low-degree melts; Gale et al., 2014, 2013), could indicate that these melts lie on a trajectory of increasing [Cu] with increasing degrees of partial melting, and consequently that melting took place in the presence of residual sulfide (see revised Fig. 1a).

However, we note (lines 253-255, page 11) that, because OPB are generated by higher degrees of partial melting than typical MORB (i.e., 30 % compared to ~20 %), OPB would be expected to have higher [Cu] if both contained residual sulfide in their mantle source. Alternatively, if, unlike MORB, OPB had exhausted sulfide during melting, the Kroenke basalts should be offset to lower [Cu] than parental-MORB, because the CSE content of the melt would be diluted by continued melting in the absence of residual sulfide. Therefore, an alternative explanation for the comparable [Cu] between primitive OPB and parental-MORB could be that sulfide is exhausted from the mantle source in both settings, and that melting terminates at the point of sulfide exhaustion. Indeed, because sulfide is suggested to become exhausted at similar degrees of melting to clinopyroxene exhaustion (from Ti and Ce systematics; Nielsen et al., 2014), this hypothesis can explain the similar [CaO] between Kroenke and primitive (~10 wt. %) MORB (see SI Fig. 1c) as noted in lines 258-260 on page 11.

We have additionally discussed PGE data for MORB and OPB to place further constraints on S-systematics during partial melting (Fig. 7). We include data for komatiites, which, because of the high degrees of partial melting associated with their formation, are typically considered to exhaust sulfide from their mantle source (komatiite data from Puchtel and Humayan, 2001). Primitive OPB (Kroenke) have near-identical CSE patterns (with the exception of Pt, Pd, Re and Au) to parental-MORB, which, together with their chondritic S/Se, suggest that neither the OPB or MORB mantle source contains residual sulfide. The slight differences between their [Pt], [Pd], [Re] and [Au] can be explained by sulfide fractionation from MORB during crustal differentiation (i.e., CSE with the highest $D^{\text{sulf/sil}}$ are most sensitive to sulfide fractionation). By contrast, sulfide-under-saturated komatiites (i.e., those which have not become saturated during crustal differentiation) are offset to lower [Pt], [Pd] and [Cu] than OPB. This is consistent with the modelling of Ding and Dasgupta (2018) which predicts that the [CSE] of partial melts will decrease by continued melting in the absence of residual sulfide. Hence, the comparable CSE patterns of MORB and primitive OPB and the slightly higher [Pt] and [Pd] of the Kroenke basalts relative to komatiites suggests that both MORB and OPB have largely exhausted sulfides in their mantle source, and furthermore that melting terminated close to the point of sulfide-exhaustion in both settings. OPB mantle source estimates,

derived assuming that sulfide was exhausted from the mantle source and that OPB are generated by 30 % partial melting, are in very good agreement to the primitive mantle estimates of McDonough and Sun (1995) and close to the estimates of Palme and O'Neill (2013), further supporting our updated interpretation that sulfide was exhausted from the OPB mantle source (lines 281-283, page 12), but that melting terminated at this point (i.e., dilution of the CSE during continued melting was minimal).

Importantly, we highlight that because both OPB and MORB appear to have stopped melting at the point of sulfide and clinopyroxene exhaustion (i.e., primitive OPB and parental-MORB have similar [CaO]), yet OPB are generated by higher degrees of melting, these systematics require that the OPB mantle source must have been more fertile (261-263, page 11), which is in agreement with general consensus for plume-related magmas. We have emphasised this point by including mantle source estimates of the CSE and S for OPB and MORB (discussed below and in lines 307-315, page 13).

3) The last but not the least, if the primitive OPBs with ~900 ppm S derived by ~30% degree of melting is indeed in equilibrium with sulfide in its mantle source, as the authors suggest, it indicates > 270 ppm S present in its mantle source, which is much higher than most of the current estimate of S abundances in the upper mantle. This would be a very interesting conclusion showing huge sulfide heterogeneity in the Earth mantle. However, it seems not convincing enough to come up with this conclusion solely based on limited numbers of high MgO samples from a single sample place (Kroenke).

Following our comments above, we believe that we have strengthened our discussion regarding the possibility that sulfide was largely exhausted from the OPB and MORB mantle sources. Assuming that S behaved incompatibly when sulfide was exhausted and that OPB were generated by 30 % melting, we can derive a minimum estimate of ~247 ppm for the OPB mantle source. Similarly, an estimate of 177 ppm can be derived assuming 20 % melting of the MORB-source mantle (lines 307-315, page 13). The S estimate of the OPB-source mantle is slightly higher than the estimate of Palme and O'Neill (2013) but is comparable to the estimate of McDonough and Sun (1995), further supporting our interpretation that melting terminated close to the point of sulfide exhaustion. Because [S] is diluted by continued melting in the absence of sulfide, we have been careful to stress that these are strictly minimum estimates (line 311, page 13).

4) Since the fate of sulfide during partial melting and the partitioning of chalcophile elements (Cu, Ag) are closely related to the pressure, temperature, degree of melting, and melt composition. I would recommend the authors to conduct some forward calculations along the melting path at the appropriate mantle potential temperature related to OPBs modelling the change of S, Cu and Ag in the partial melts. And if indeed the Cu/Ag ratio can be reproduced with an elevated sulfur concentration in the mantle, it would be great for the authors to also compile chalcophile element data from some other locations which are plume related and derived by high degree melting to check if such sulfur enrichment is just a local effect, or is a common phenomenon.

Cu/Ag is not fractionated by sulfide melt (Jenner, 2017). Hence, increasing the S in the mantle (or proportion of sulfide melt) will not change the modelled Cu/Ag. Only melting in the

absence of sulfide and/or in the presence of crystalline sulfide will fractionate Cu/Ag. Detailed forward modelling of CSE behaviour during partial melting was tackled in the manuscripts by Ding and Dasgupta (2018; 2017). Ding and Dasgupta demonstrate that modelling the behaviour of S during mantle melting processes is highly sensitive to variables such as the initial choice of [CSE] abundances, major element modelling (i.e., source mineralogy and fertility, depth and temperature of melting), composition of the sulfide and even the choice of model used to calculate the sulfide content at sulfide saturation (e.g., comparisons on Figs. 2, 6, 7, 8 & 10 of Ding and Dasgupta, 2017 and Figs. 4, 5, 6, 7 & 8 of Ding and Dasgupta, 2018). Resolving these issues is beyond the scope of our manuscript. Hence, we prefer to stick to a simple finding that has been replicated by numerous experimental studies: the [Cu] of a melt increases with increasing F until sulfide becomes exhausted in the source. Consequently, we decided to use the compositions of low-degree melts (i.e., the reviewer states that these should contain residual sulfide), rather than high degree OIB melts, to strengthen our discussions. Unlike plateaus, many OIBs (e.g., Hawaii and Iceland) degas S during subaerial eruption and/or Cu/Ag data is unavailable (i.e., much of the published Ag data lacks the necessary ZrO interference correction). Furthermore, Ding and Dasgupta (2018) recently suggested that recycled crustal components influence the S-systematics of OIB. Given this complexity, we do not find it suitable to draw comparison between other plume-related settings and OPB until further research is undertaken to constrain the effects of source heterogeneity on chalcophile element systematics. Instead, we use the compositions of komatiites which fit the predictions of the Ding et al modelling (i.e., [Cu] that are lower than primitive MORB).

5) Another problem I have with the current manuscript is that even though all the new OPBs data cover a wide range of MgO from >10 wt.% to 5 wt.%, samples from individual OPB suite are either within small MgO range or with quite distinct MgO concentration. Since these different OPB suites are not co-genetic, more clarification and justification is needed to show that samples from each suite are indeed representative for the whole differentiation trend. For example, the conclusion that the OPBs are sulfide-saturated through differentiation mainly results from the slightly negative slopes of the highly chalcophile elements. However, it is not clear only through the method part that whether the calculated slopes in Figure 4 are representative of the whole differentiation range between 10-5 wt.% MgO for all the OPB suites, or are derived from a certain OPB suite at a certain MgO range. Since one of the key discussions of this manuscript is that why OPBs generally hit sulfide-saturation later than MORBs, it is important to clarify for individual OPB suite at what MgO content the basalts become sulfide saturated during crystallization.

We would first like to emphasise, the use of the bulk-partitioning arguments (slopes) is one example of data-handling that we used to demonstrate that the OPB samples reached sulfide-saturation. We also used a comparison between the Ori and the Tamu samples (Fig. 5), we include a laser scan whereby a sulfide was ablated during analysis (SI Fig. 4) and we refer to the literature whereby sulfides have been observed in samples. We also focus on the use of element ratios (e.g., the increasing S/Se of the Ori samples). Hence, we are not basing our conclusions entirely on the 'slightly negative slopes of the highly chalcophile elements'. However, we would like to emphasise that the derived relative bulk-partitioning information (e.g., Rb is more incompatible than the REE) is consistent with expected bulk Ds (see modelling in O'Neill and Jenner, 2012).

It is stated in the text that the slopes are calculated using 'OPB slopes are calculated using only the most-evolved Kroenke samples (where plagioclase is inferred to join the liquidus, see Fig. 1 and SI) and the lower [MgO] OPB suites (except the S-degassed Tamu Massif) to allow comparison between MORB and OPB. A second slope (Fig. 5) is calculated for each element following the above method, but excluding two samples that have extremely high incompatible trace element contents (e.g., Rb, Cs and Th) at a given [MgO] compared to the other OPB samples. The slopes calculated with and without the outliers (Fig. 5) are within error and/or are indistinguishable for elements more compatible than La, but diverge slightly for the most incompatible elements (e.g., U and Th). We are also quite careful in the text to emphasise that 'Although the OPB suites are not expected to be co-genetic, these systematics suggest that OPB magmas have similar liquid lines of descent to MORB (i.e., fractionated similar mineral assemblages and consequently, equilibrated at similar crustal pressures prior to eruption) between 9-6 wt. % MgO'. We have also reworded the section emphasising that the 'the Kwaimbaita and Ori Massif magmas appear to reach sulfide saturation at ~7.5 wt. % MgO' and 'the Cu systematics of lower [MgO] Kerguelen samples indicate sulfide-saturation at ≤ 6.5 wt.% MgO'.

We have updated the Supplementary Information to give an extended discussion of the use of slopes. Despite covering distinct ranges in [MgO], each OPB suite plots close to the predicted 0.1 GPa fractional crystallisation trend (i.e., liquid line of descent), modelled using MELTS for the most primitive (highest [MgO]) Kroenke sample. Importantly, this indicates that the OPB equilibrated at similar pressures in the crust prior to eruption and that the Kroenke basalts are a reasonable approximation for the parental melt composition (major elements) of all the OPB suites. Furthermore, the gradients of each individual OPB suite (except for the 'olivine-only' Kroenke) on, for example [FeO_T] or [Al₂O₃] versus [MgO] plots (SI figure 1a, b) are similar, indicating that OPB fractionated similar mineral assemblages during crustal differentiation.

More detailed comments

Line 69-71: This sentence seems to be misleading. At higher pressure, sulfide solubility would be lower, then one would expect that it takes less crystallization to reach sulfide saturation rather than more pronounced crystallization.

Our initial intention was to state that OPB melts were generated at higher pressures than MORB (i.e., under thicker crust) and therefore had lower initial [S] because of the pressure effects on [S]_{SULF-SAT}. The melts would therefore require more fractional crystallisation during differentiation in the upper crust to become sulfide-saturated (i.e., if the pressure of differentiation was the same as MORB). We have, however, amended this paragraph to outline the additional conclusions we have reached by considering new modelling and data during revision (e.g., PGE data).

Line 83: what is calculated? How?

Parental-MORB values are taken from Jenner (2017) and are not calculated independently in this study with the exception of Se, Mo and Sn which are calculated for the new MORB data we have obtained (using the methods detailed in Jenner, 2017). We clarify this in Methods.

Line 91: If one looks at Cu and Ag systematics for individual suites in Figure 1, only samples from Tamu suite show increasing trend.

We were initially trying to state that when OPB suites are considered together, they show increasing [Cu] and [Ag] with decreasing [MgO]. We have modified this sentence and it now reads 'Together, the OPB suites show increasing [Cu] and [Ag] with decreasing [MgO] (Fig. 1) and have lower [S] at a given [FeO_T] compared to the MORB array (Fig. 2)'.

Line 92: In Figure 2, the sulfur concentration of all the OPBs data is about 100-200 ppm lower than S in the MORBs. Therefore, the word "significant" seems to be overused.

Sentence has been revised and now reads 'and have lower [S] at a given [FeO_T] compared to the MORB array (Fig. 2)'.

Line 174-175: How much S would be present if this is the case? How would the estimate compare to the S abundances of other mantle domains producing high-degree mantle melts?

See discussion above.

Line 217-218: If the Kroenke primitive melts are also sulfide saturated as MORBs, why would the [Re], [Pt] and [Au] be the closest estimates of the unmodified mantle melts?

Most MORB have fractionated sulfide melt and consequently, they are depleted in Re, Pt and Au. Based on the high [Pt] of the Kroenke relative to sulfide-saturated MORB, we infer that this suite is sulfide-under-saturated. Our suggestion is that, because these samples are not sulfide-saturated, their [Pt], [Re] and [Au] might be reasonable estimates of the contents of these elements in primitive mantle-derived melts which have not become sulfide-saturated during crustal differentiation (i.e., they ascended through the mantle too quickly to become sulfide-saturated).

--

Final comments: We thank the reviewer for their detailed and constructive comments and we are glad that they find the topic applicable to a wide range of fields. By embracing their comments and criticisms, we believe that we have considerably strengthened the arguments in our manuscript, especially regarding whether or not sulfide is a residual phase in the mantle during the formation of oceanic plateaus and MORB.

Reviewer #3: (Anonymous)

The main point of the article is that for a given [MgO], OPB have lower CSE contents than would be expected if these elements were acting perfectly incompatibly (or close to it in the case of copper). This suggests that they must be sulfide saturated, although this happens later

than in MORB, leading to and enrichment in CSE. The article suggests that the later saturation is a reflection of a larger pressure difference (and therefore sulfur solubility difference) between melting and subsequent differentiation of melts. Finally they suggest that degassed OPB samples have higher CSE related to resorption of sulfides.

Major comments

1) My main criticisms are related to the lack of uncertainties. There don't appear to be error bars on any of the geochemical plots, and uncertainties on individual analyses would be useful in the supplementary table. This is particularly important for plots of the Ag and Se content, where uncertainties are likely high, and for the ratios in Figure 3 (particularly 3a, since Se and S data come from different sources with individual uncertainties).

We had initially reported the errors associated with our analyses in the Methods section of our manuscript and supplementary tables 1 and 2. However, we are in agreement with the reviewer that the inclusion of error bars, especially for low concentrations Se and Ag analyses, is important for effectively interpreting the data. We have therefore amended our figures accordingly to include error bars on Se and Ag data and propagated errors on S/Se, Cu/Ag and S*. Error bars given for LA-ICP-MS typically reflect precision (repeat analyses) rather than accuracy. Hence, we report errors from repeat analysis of the NWLSC standard (see Methods). These standards are characterised repeatedly during each analytical session to estimate spectral interferences on Se and Ag and therefore provide the most statistically relevant error associated with the analyses of these elements.

Uncertainties for other elements are reported from repeat analyses of the BCR-2G standard and given in Supplementary Table 1 along with comparisons with literature data (accuracy). Additionally, we now include our new MORB data for Se (collected on the same LA-ICP-MS system and using the same interference corrections) to ensure that comparisons between the MORB and OPB arrays are as reliable as possible.

2) To some extent this affects the conclusions. Firstly the authors suggest that Tamu samples must have been sulfide saturated because S values are high and that apparently D=0 behavior is caused by resorption of sulfides. However S* values are calculated based on S/Se ratios and the Tamu S contents, both of which have uncertainties which must be propagated through and appropriate error bars for S* need to be plotted. This conclusion is also based on the assumption that Ori S/Se ratio are a good estimate for Tamu which may be true, although there appears to be significant spread in the S/Se ratios at other OPB in this study. It would be interesting to see a range of S* values based on the range of S/Se ratios.*

We have added propagated errors to S/Se and S* estimations following the reviewer's suggestions in figures 1, 2 and 3. Accounting for this error, S* estimations plot below the modelled 'D=0' trajectories, and therefore suggest that the Tamu Massif samples may have been sulfide-saturated prior to S-degassing. Only the average S/Se of the compositionally similar (i.e., low [MgO] Ori sample group) Ori Massif samples, which span a similar [MgO] to the Tamu Massif samples, are used to estimate the S* of the Tamu samples. Ranges in S*, based on calculated errors, are shown on figures 1 and 2. The range in S/Se of the various

OPB magmas supports our conclusions that the more evolved OPB magmas fractionated sulfide (i.e., the MORB array shows an increase in S/Se with decreasing MgO).

3) Secondly the assertion that sulfide exhaustion can't explain lower S contents and explain later sulfide saturation is based on the fact that Cu/Ag ratios aren't significantly offset from MORB values. It would be really useful to see a back of envelope calculation here. Authors could calculate roughly what decrease in the sulfur contents of primary melts would be needed (maybe a few 100 ppm from the FeO_T vs S plot) to explain their results. They could then calculate how much partial melting would have to occur after sulfide exhaustion to get these lower S contents, and what the expected difference in the Cu/Ag ratio would be, this could then be added to Figure 3b.

In our initial submission, we argued that, because the Cu/Ag of OPB are not significantly offset from MORB (i.e., the majority of samples are within error of the MORB array), OPB must contain residual sulfide in their mantle source, and consequently that primitive OPB magmas were sulfide-saturated prior to ascent and crustal differentiation. This assessment was based on the modelling results presented by Li and Audétat (2012), who showed that at the point of sulfide exhaustion, Cu/Ag decrease to ~0 almost instantaneously with increasing degrees of melting. We would like to extend this modelling to include the effect of the partitioning of Cu and Ag into silicates following sulfide exhaustion, as the reviewer suggests. However, as we highlight in the revision, the accuracy/availability of Ag partitioning data is not of sufficient quality. Instead, we have extended our discussion of S-systematics during partial melting as outlined above (see response to Reviewer 2). In addition, in the Section titled 'Crustal Processes and CSE systematics' we now estimate how low the S of the Tamu Massif melts at 9 wt.% would need to be (400 ppm) to explain these melts not attaining S saturation during differentiation.

4) In figure 2 it wasn't clear why the modeled [S]_{sulf-sat} isobars for plotted for OPB's and for MORB had such a different shape. The text suggests the OPB isobars are linear fits but it wasn't clear on what data they were linear fits to.

We have adjusted the text (lines 326, page 14) to clarify that isobars are modelled from individual [S]_{SULF-SAT} data points, calculated for a given glass composition at a specified pressure. The different shape and shallower slopes of the OPB isobars (in Fig. 2) is because of compositional effects on [S]_{SULF-SAT}. Specifically, because OPB reach sulfide-saturation at lower [MgO] than MORB, [Cu] and [Ni] first increase relative to MORB. This has the effect of suppressing [S]_{SULF-SAT} (Smythe et al., 2017) and hence leads to isobars which diverge from MORB with increasing [FeO_T] (see Fig. 2).

5) Finally I think it would be useful to have some of the plots used to calculate slopes for bulk-D values in the supplementary data. Looking at the spread in the MgO vs Ag data (and with associated uncertainties on the Ag values) it seems like it should be difficult to produce a fit which is statistically different from the trajectory of incompatibility.

We agree with the reviewer that clarification of how slopes are calculated would be useful to the reader. We have therefore included an extended discussion of how slopes are calculated, including which samples are excluded from slope calculations, in the Supplementary

Information. In revision, we have calculated a second slope for each trace element. This addition is detailed in lines 145-149 on page 6. This second slope is calculated following the same method to our initial slopes (i.e., excluding 'olivine-only' Kroenke and S-degassed Tamu Massif samples) but additionally excludes two samples (one Ori Massif and one Kerguelen) which are outliers on incompatible trace element (e.g., Rb, Cs and Th) versus [MgO]. Inclusion of both slopes therefore provides an estimate of the range in compatibility of each trace element during OPB crustal differentiation, rather than assuming that all suites would plot along one line. Importantly, the slopes for each trace are within error except for a few of the most incompatible elements (e.g., Nb on figure 5).

As correctly highlighted by Reviewer 3, it is difficult to establish a statistical fit through the CSE. This a consequence of each of the CSE reaching sulfide-saturation at different [MgO] which leads to considerable scatter when all samples are considered as a singular OPB 'array'. However, in all cases where some of the samples are sulfide-saturated (i.e., offset from 'D=0' behaviour), the slope of the element will be skewed to more positive values (i.e., a more positive bulk-D). Hence, the slope of each CSE does not reflect its true compatibility in each OPB suite but instead shows that, as a whole, some of the OPB samples were sulfide-saturated because slopes are offset to more positive values than the slopes of highly incompatible elements (e.g., Rb).

I think that the conclusion that OPB are sulfide saturated, but that this occurs later than in MORB is likely valid, based on the data presented in Figure 1. The arguments made for the other claims in the paper make sense but are hard to asses without uncertainties on the data

We are happy that the reviewer concludes that our conclusions are valid, and our arguments make sense to them. In our revised manuscript, we have followed the reviewer's suggestion to provide uncertainties on our data and we consider that this has substantially improved our manuscript and strengthened its central argument.

References

- Becker, H., Horan, M.F., Walker, R.J., Gao, S., Lorand, J.P., Rudnick, R.L., 2006. Highly siderophile element composition of the Earth's primitive upper mantle: Constraints from new data on peridotite massifs and xenoliths. *Geochim. Cosmochim. Acta* 70, 4528–4550.
- Ding, S., Dasgupta, R., 2018. Sulfur inventory of ocean island basalt source regions constrained by modeling the fate of sulfide during decompression melting of a heterogeneous mantle. *J. Petrol.* (in press).
- Ding, S., Dasgupta, R., 2017. The fate of sulfide during decompression melting of peridotite – Implications for sulfur inventory of the MORB-source depleted upper mantle. *Earth Planet. Sci. Lett.* 459, 183–195.
- Gale, A., Dalton, C.A., Langmuir, C.H., Su, Y., Schilling, J.G., 2013. The mean composition of ocean ridge basalts. *Geochemistry, Geophys. Geosystems* 14, 489–518.
- Gale, A., Langmuir, C.H., Dalton, C.A., 2014. The global systematics of ocean ridge basalts and their origin. *J. Petrol.* 55, 1051–1082.
- Jenner, F.E., 2017. Cumulate causes for the low contents of sulfide-loving elements in the continental crust. *Nat. Geosci.* 10, 524–529.

- Li, Y., Audétat, A., 2012. Partitioning of V, Mn, Co, Ni, Cu, Zn, As, Mo, Ag, Sn, Sb, W, Au, Pb, and Bi between sulfide phases and hydrous basanite melt at upper mantle conditions. *Earth Planet. Sci. Lett.* 356, 327–340.
- McDonough, W.F., Sun, S. -s., 1995. The composition of the Earth. *Chem. Geol.* 120, 223-253.
- Nielsen, S.G., Shimizu, N., Lee, C.-T. a., Behn, M.D., 2014. Chalcophile behavior of thallium during MORB melting and implications for the sulfur content of the mantle. *Geochemistry Geophys. Geosystems* 15, 4905–4919.
- Palme, H., O'Neill, H. St. C., 2013. *Cosmochemical Estimates of Mantle Composition*, 2nd ed, *Treatise on Geochemistry: Second Edition*. Elsevier Ltd.
- Puchtel, I. S., and Humayan, M., 2001. Platinum group element fractionation in a komatite basalt lava lake. *Contrib. to Mineral. Petrol.* 65, 2979–2993.
- Roberge, J., Wallace, P.J., White, R. V., Coffin, M.F., 2005. Anomalous uplift and subsidence of the Ontong Java Plateau inferred from CO₂ contents of submarine basaltic glasses. *Geology* 33, 501–504.
- Smythe, D.J., Wood, B.J., Kiseeva, E.S., 2017. The S content of silicate melts at sulfide saturation: New experiments and a model incorporating the effects of sulfide composition. *Am. Mineral.* 102, 795–803.

Reviewer #1 (Remarks to the Author):

I have read the revised version of this nice and interesting contribution. In my opinion the authors have carefully and correctly addressed the criticisms/questions raised by the referees. I have no further comments to do and recommend acceptance of the present manuscript.

Bruno Scaillet

Reviewer #2 (Remarks to the Author):

The revised manuscript "Pressure effects complicate our understanding of Earth's sulfur cycle" by Reekie et al. has reported analyses of major and trace elements in several oceanic plateau basalts and described the sulfur and chalcophile-siderophile elements systematics during partial melting and fractional crystallization. The revised the manuscript addressed the comments from previous reviewers' well, and extended the discussion on the degassing and partial melting, as well as estimate the abundances of trace elements in their mantle source. These additions strengthen the idea of the pressure/crustal thickness effects on the CSE systematics during crystallization in oceanic plateau basalts and enhance the potential impacts of the manuscript. I only have a few minor comments on this revised manuscript. I recommend the manuscript to be published in Nature Communications.

1. The authors interpreted the significantly lower S/Se of Tamu samples as S degassed; however, some of the Tamu samples also show very high H₂O contents in Figure 4a, even higher than the fractional crystallization trend defined by Figure 4a. Is there any explanation of this abnormally high H₂O contents?
2. The authors used the sulfide-saturated MORBs systematics to derive the partitioning behavior of Cu and Ag during differentiation (line 175-177, Figure 1a, b) and applied the same slopes to the analysis of oceanic plateau basalts. This estimate essentially assumes that there is no effect of pressure, temperature and melt composition on the bulk partitioning behavior of Cu and Ag, which is a reasonable assumption. However, previous experiments have shown that Cu partition coefficient between sulfide and melt is affected by temperature and melt composition (Kiseeva and Wood, 2015). Given the crystallization pressure differences between MORBs and oceanic plateau, basalts can be 0.1 GPa vs. 1 Gpa, one would expect the temperature, and melt composition is also different. It is worth mentioning that it is assumed that no T, P, and melt composition effect on the partitioning between sulfide and melt in the text.
3. Different symbols are used for the same location in different figures, which is kind of confusing. I suggest the authors use the same symbols for one location in all the figures, if possible.

Reviewer #3 (Remarks to the Author):

In this revision the authors have addressed my concerns regarding uncertainties and I believe the data presented supports the conclusions that OPB are often sulfide saturated and that resorption of sulfides (at least in the case investigated) can lead to enrichment in CSE.

Although it is far from the sole focus of the paper, I think that the manuscript is particularly strengthened by a more thorough discussion of sulfide saturation during mantle melting.

Reviewer Comments

Editor:

Your manuscript entitled "Pressure effects complicate our understanding of Earth's sulfur cycle" has now been seen again by our referees, whose comments appear below. In light of their advice I am delighted to say that we are happy, in principle, to publish a suitably revised version in Nature Communications under the open access CC BY license (Creative Commons Attribution v4.0 International License).

We therefore invite you to revise your paper one last time to address the remaining concerns of our reviewers.

We appreciate the editor's and reviewer's detailed and constructive comments which have contributed to improving the quality of our submission. The remaining comments and concerns of the reviewers are addressed below and we have revised our manuscript where necessary.

Reviewer #1: (Bruno Scaillet)

I have read the revised version of this nice and interesting contribution. In my opinion the authors have carefully and correctly addressed the criticisms/questions raised by the referees. I have no further comments to do and recommend acceptance of the present manuscript.

We appreciate that the reviewer finds our contribution of interest. We thank the reviewer for their constructive comments and we are pleased that they recommend acceptance of our manuscript.

Reviewer #2: (Anonymous)

The revised manuscript "Pressure effects complicate our understanding of Earth's sulfur cycle" by Reekie et al. has reported analyses of major and trace elements in several oceanic plateau basalts and described the sulfur and chalcophile-siderophile elements systematics during partial melting and fractional crystallization. The revised the manuscript addressed the comments from previous reviewers' well, and extended the discussion on the degassing and partial melting, as well as estimate the abundances of trace elements in their mantle source. These additions strengthen the idea of the pressure/crustal thickness effects on the CSE systematics during crystallization in oceanic plateau basalts and enhance the potential impacts of the manuscript. I only have a few minor comments on this revised manuscript. I recommend the manuscript to be published in Nature Communications.

Comments

1) The authors interpreted the significantly lower S/Se of Tamu samples as S degassed; however, some of the Tamu samples also show very high H₂O contents in Figure 4a, even

higher than the fractional crystallization trend defined by Figure 4a. Is there any explanation of this abnormally high H₂O contents?

The majority of Tamu (and Ori) samples follow a liquid line of descent which indicates that they are H₂O-under-saturated. Given that H₂O degassing will lead to lower [H₂O] in degassed melts, it is likely that the Tamu and Ori Massif samples with anomalously high [H₂O] are also H₂O-under-saturated. As per the experimental results of Dixon et al. (1995), the higher [H₂O] of tholeiitic liquids at a given stage of differentiation might suggest that melts were fractionated and/or erupted at higher pressure. However, this pressure would have to be low enough, in the case of the anomalous Tamu Massif samples, that S could still degas.

Given that an estimation of eruption depths based on volatile element contents is model dependent and the depth at which the samples were obtained is not a true reflection of their eruption depth (because of post-eruption subsidence), an exact assessment of the H₂O- and CO₂-systematics of the melts at eruption is challenging. Instead, we choose to focus on the simple observation that, unlike for S, the Tamu samples are H₂O-under-saturated.

2) The authors used the sulfide-saturated MORBs systematics to derive the partitioning behavior of Cu and Ag during differentiation (line 175-177, Figure 1a, b) and applied the same slopes to the analysis of oceanic plateau basalts. This estimate essentially assumes that there is no effect of pressure, temperature and melt composition on the bulk partitioning behavior of Cu and Ag, which is a reasonable assumption. However, previous experiments have shown that Cu partition coefficient between sulfide and melt is affected by temperature and melt composition (Kiseeva and Wood, 2015). Given the crystallization pressure differences between MORBs and oceanic plateau, basalts can be 0.1 GPa vs. 1 GPa, one would expect the temperature, and melt composition is also different. It is worth mentioning that it is assumed that no T, P, and melt composition effect on the partitioning between sulfide and melt in the text.

As correctly stated by the reviewer, we have chosen to estimate the bulk-partitioning behaviour of Cu and Ag at sulfide-saturation (red-dashed arrows on figure 1) from the systematics of these elements in MORB, which are sulfide-saturated. This approximation assumes that there is no effect of pressure, temperature and melt composition on the partitioning behaviour of Cu and Ag between sulfide melt and silicate melt. We also note that this approximation does not account for differences in the amount of sulfide that may be fractionating in each melt. There is also no experimental parameterisation for the effect of pressure on $D^{\text{sulf/sil}}$ values. Given this uncertainty, we have chosen to provide an estimate of the partitioning behaviour of Cu and Ag at sulfide-saturation as an approximation only. We have clarified this point in the manuscript by stating that “The partitioning behaviour of Cu and Ag after the melts reach sulfide-saturation can be approximated from the slopes of these elements in sulfide-saturated MORB (red dashed arrows on Fig. 1a, b), assuming for simplicity that there is no effect of temperature, pressure and melt composition on $D^{\text{sulf/sil}}$ ” (lines 176-179).

3) Different symbols are used for the same location in different figures, which is kind of confusing. I suggest the authors use the same symbols for one location in all the figures, if possible.

We speculate that the reviewer is referring to the symbols used in figure 7 and its comparison to figures 1-5. We have therefore modified figure 7 accordingly to avoid any possible confusion that may arise.

--

Final comments: We thank the review for the detailed and constructive comments which have significantly improved the quality of our manuscript. We are extremely pleased that they recommend acceptance for publication in Nature Communications.

Reviewer #3: (Anonymous)

In this revision the authors have addressed my concerns regarding uncertainties and I believe the data presented supports the conclusions that OPB are often sulfide saturated and that resorption of sulfides (at least in the case investigated) can lead to enrichment in CSE

Although it is far from the sole focus of the paper, I think that the manuscript is particularly strengthened by a more thorough discussion of sulfide saturation during mantle melting.

We are pleased that the reviewer supports our conclusions for the presented data and that the manuscript has been strengthened by a discussion of sulfide-systematics during mantle melting. We thank the reviewer for taking the time to review our submissions and for their constructive comments which have improved the quality of our manuscript.

References

- Dixon, J.E., Stolper, E.M., Holloway, J.R., 1995. An experimental study of water and carbon dioxide solubilities in mid-ocean ridge basaltic liquids. Part I: Calibration and solubility models. *J. Petrol.* 36, 1607-1631.
- Kiseeva, E.S., Wood, B.J. 2015. The effects of composition and temperature on chalcophile and lithophile element partitioning into magmatic sulphides. *Earth Planet. Sci. Lett.* 424, 280-294.